# Tracking the relation between gist and item memory over the course of long-term memory consolidation

**Tima Zeng\*, Alexa Tompary, Anna C Schapiro, Sharon L Thompson-Schill**

Department of Psychology, University of Pennsylvania, Philadelphia, United States

**Abstract** Our experiences in the world support memories not only of specific episodes but also of the generalities (the 'gist') across related experiences. It remains unclear how these two types of memories evolve and influence one another over time. In two experiments, 173 human participants encoded spatial locations from a distribution and reported both item memory (specific locations) and gist memory (center for the locations) across 1–2 months. Experiment 1 demonstrated that after 1 month, gist memory was preserved relative to item memory, despite a persistent positive correlation between them. Critically, item memories were biased toward the gist over time. Experiment 2 showed that a spatial outlier item changed this relationship and that the extraction of gist is sensitive to the regularities of items. Our results suggest that the gist starts to guide item memories over longer durations as their relative strengths change.

## Introduction

Our experiences in the world are perceived and remembered both as individual items, events, and episodes, and also as aggregated collections or sets of related items with common properties. For example, one can remember seeing a brown bear at the zoo, a polar bear at an aquarium, an animated bear in a Winnie the Pooh movie, and on and on; but one also can readily understand the phrase 'smarter than your average bear' by aggregating over those individual experiences. A fundamental question in cognitive science is how we extract summary statistics from individual instances, both during perception and working memory (where aggregated information is often referred to as an 'ensemble') and during episodic encoding and retrieval (where aggregated information is often referred to as a 'schema' or as the 'gist' of an experience) [The 'gist' in this paper means 'generalities across individual instances', instead of 'lack of details', 'less precise', or 'abstract' as in fuzzy trace theory (*Brainerd and Reyna, 2002*)]. In addition, researchers have attempted to characterize how memory for these types of information changes over time. For example, studies of long-term memory in both humans and animals have demonstrated that gist memory persists or even improves over time, whereas memory for the individual items from which the gist is built fades (*Posner and Keele, 1970*; *Richards et al., 2014*).

What do these observations of temporal dissociations tell us about the relation between item memory and gist memory? On the one hand, a persisting gist memory with less accurate item memory is often taken as evidence that a gist representation becomes independent of individual item representations as it is abstracted during encoding (*Posner and Keele, 1970*) or through consolidation (*Richards et al., 2014*). On the other hand, a persisting gist memory with less accurate item memory is not sufficient evidence for the independence of a gist representation: Even when item memories become noisy and less accurate, they still can retain enough information to support a relatively intact memory of gist at retrieval (*Alvarez, 2011*; *Squire et al., 2015*).

Disentangling these two possibilities based on existing evidence is difficult, because previous studies do not have a direct measurement of the gist information retained in item memories. In this

\*For correspondence:
zengtima@gmail.com

Competing interest: See
page 21

Reviewing editor: Muireann
Irish, University of Sydney,
Australia

study, we developed a paradigm to test item memory, gist memory, and 'estimated' gist memory, which is an estimate of gist memory given the assumption that it is assembled from individual memories of constituent items. Ensemble perception research was a source of inspiration in developing such a paradigm. Studies of rapid perception of complex visual arrays reveal precise representations of gist (i.e. ensemble statistics) with less accurate item memory retrieval in working memory (*Ariely, 2001*). In order to investigate the relation between item and gist, ensemble perception paradigms often operationalize the gist as the average representation across instances.

Following this reasoning, we operationalize item memory as a set of landmarks (e.g. restaurant and university) whose locations are clustered together on a screen, and gist memory as the center for these landmark locations. In our paradigm, participants learn this set of landmarks, and are then asked to report the spatial center of these landmarks and recall the location of each landmark. Importantly, an 'estimated center' can be computed based on their retrieval of individual items, and its accuracy can thus reveal the amount of gist information available in item memories. Thus, we can investigate the relation between gist memories and item memories over the course of long-term memory consolidation by measuring the relationship between the accuracy of the reported center and the estimated center. A positive correlation between estimated and reported center accuracy could mean that participants' gist memory was still supported by individual item memories, or that the gist was influencing the retrieval of items.

To probe the direction of this relationship, we developed a gist-based bias measurement, an approach borrowed from research on hierarchical clustering models and from semantic memory, which both reveal how gist memory influences memory for specific items (*Brady and Alvarez, 2011*; *Hemmer and Steyvers, 2009*; *Huttenlocher et al., 1991*; *Tompary and Thompson-schill, 2019*). This measure of bias indicates the magnitude of the particular direction in which the items were attracted. Theories suggest that this influence reveals a reconstructive memory retrieval process (*Brady et al., 2015*; *Hemmer and Steyvers, 2009*; *Schacter et al., 2011*) that depends on the relative strength of item and gist memory (*Tompary et al., 2020*). Consistent with this theory, prior work in long-term memory consolidation, which examines gist memory that is newly acquired, has shown that over time, as the strength of gist memory is preserved or improves and/or that of item memory decreases, items that are consistent with the gist are recalled more precisely (*Richter et al., 2019*; *Sweegers and Talamini, 2014*; *Tompary et al., 2020*). However, these results did not demonstrate a gist-based bias, a distortion of item memory from such a newly acquired gist. An increasing bias of item memories toward the remembered gist — in this paradigm, the reported center — would be strong evidence for the increasing influence of gist memory over item memories. Our interest in examining the influence of gist memory on item memory at long delays stems from a desire to bridge the literature reviewed above with a potentially related literature reporting the effects of prior knowledge on memory retrieval (e.g. *Huttenlocher et al., 2000*; *Tompary and Thompson-schill, 2019*).

The current study aimed to understand the relation between item and gist memory over the course of a month. In Experiment 1, we trained three groups of participants on spatial locations of six landmarks, and we measured the change of error in memory ffor these items as well as the memory for the gist (i.e. the center participants reported) at one of three delay periods: 24 hr, 1 week, or 1 month. We predicted that the accuracy of the reported center would persist or improve despite the accuracy of retrieved items decreasing over a month, as seen in prior work. We extended prior observations by including two new measures—estimated center and gist-based bias—in order to explore how the relation between memories for items and the gist changes over the course of 1 month. In order to understand how influence of a gist memory on item memory changes over time, we compared observed bias from participants with bias generated under two simulations, one assuming that participants only had item memory, the other assuming that participants had item memory and a separate gist representation.

In Experiment 2, we explored the influence of an 'outlier' item in spatial location both on the gist, and on the relation between item and gist memories demonstrated in Experiment 1. Research in long-term memory (*Richards et al., 2014*) and working memory (*Whitney and Yamanashi Leib, 2018*) shows that outliers that are inconsistent with the pattern across all items differently influence the memory for the pattern, compared to items that are more consistent with the pattern. Outliers greatly disrupted or shifted the overall pattern (*Richards et al., 2014*), or were discounted in estimating the pattern (*Haberman and Whitney, 2010*) compared to other items. In Experiment 2, we

examined the extent to which the gist representation was influenced more or less by an outlier item over time. We also investigated whether bias in item memory was to the center location including or excluding the outlier item with the same simulation approach as in Experiment 1. Taken together, these experiments provide new information about how item and gist memory are consolidated over time.

## Results

### Experiment 1

Item memories were operationalized as 'landmarks' (i.e. dots associated with unique landmark names) in six locations on a laptop screen (*Figure 1*). In Session 1, 130 participants learned the locations of the landmarks individually through a training to criterion procedure (see *Figure 1* and Materials and methods for details). After training, participants were tested on item and gist memory: First, they indicated their guess about the center of the landmarks (gist memory test), and then they recalled each landmark location, without feedback and in a random order (item memory test). After 24 hours (*n* = 44), 1 week (*n* = 43), or 1 month (*n* = 43), participants returned for Session 2, during which they completed the gist memory test followed by the item memory test again. This testing order was chosen to reduce the influence of item memories on reported gist.

### Gist memory decreased less than item memory over time

We developed an error measurement for the accuracy of item and gist memory (*Figure 2a*; See Materials and methods for details). All three delay groups performed above chance on both the item memory test (compared to chance defined as the average of distance between encoded item locations and center of the screen, and compared to chance defined as the average of distance between each encoded item location and center of all encoded locations) and the gist memory test (chance defined as the distance between the center of encoded locations and center of the screen) at both Session 1 and 2 (all p <.0001; *Figure 2—figure supplement 1*). To examine how item and gist memory changed over time, we conducted a 3 (group: 24 hr, 1 week, and 1 month) X 2 (memory type: item, center) aligned ranks transformation ANOVA of the difference in error between Session 1 and Session 2 (because the data were not normally distributed). This test revealed a main effect of group, $F(2, 254) = 42.26$, p < 0.001, a main effect of memory type, $F(1, 254) = 99.36$, p < 0.001, and an interaction between group and memory type, $F(2, 254) = 23.76$, p < 0.001. This interaction reveals that the error in retrieved items increased more over time compared to error in the reported center (*Figure 2b*). Specifically, whereas each pairwise comparison between groups was significant for item memory (Mann-Whitney tests: all p's < 0.01), the only reliable group difference for gist memory was between the 24 hr and 1 month groups ($U = 685$, p = 0.026). In addition, the change in retrieval error of item locations was significantly higher than that of reported center at a delay of 24 hr (Wilcoxon signed rank tests: $Z = 2.14$, p = 0.03), 1 week ($Z = 5.79$, p < 0.001), and 1 month ($Z = 6.53$, p < 0.0001). These results showed that item memories decreased significantly more relative to gist memory over time.

### Positive relationship between item and gist memories across 1 month

To explore the relation between item and gist memory over time, we used a linear model to evaluate the effects of estimated center error, delay group, and their interaction on reported center error. We found that estimated center error significantly predicted reported center error, $SSE = 48138$, $F(1, 124) = 18.31$, p < 0.001. The effect of delay, $SSE = 4873$, $F(2, 124) = 0.93$, p = 0.40, and the interaction between the estimated center error and delay, $SSE = 11729$, $F(2, 124) = 2.23$, p = 0.11, on reported center error was not significant. These results indicate a stable relation between item and gist memory over time; our subsequent analyses will examine the source of this relation.

### Item memory retrieval biased toward gist over time

The positive correlation could indicate that participants' reported gist was still supported by individual item memories, or alternatively that the reported gist was influencing the retrieval of items. To examine the direction of the relation between item and gist memories, we developed a bias measurement (see Materials and methods for details) as an index of how much the retrieval of each item

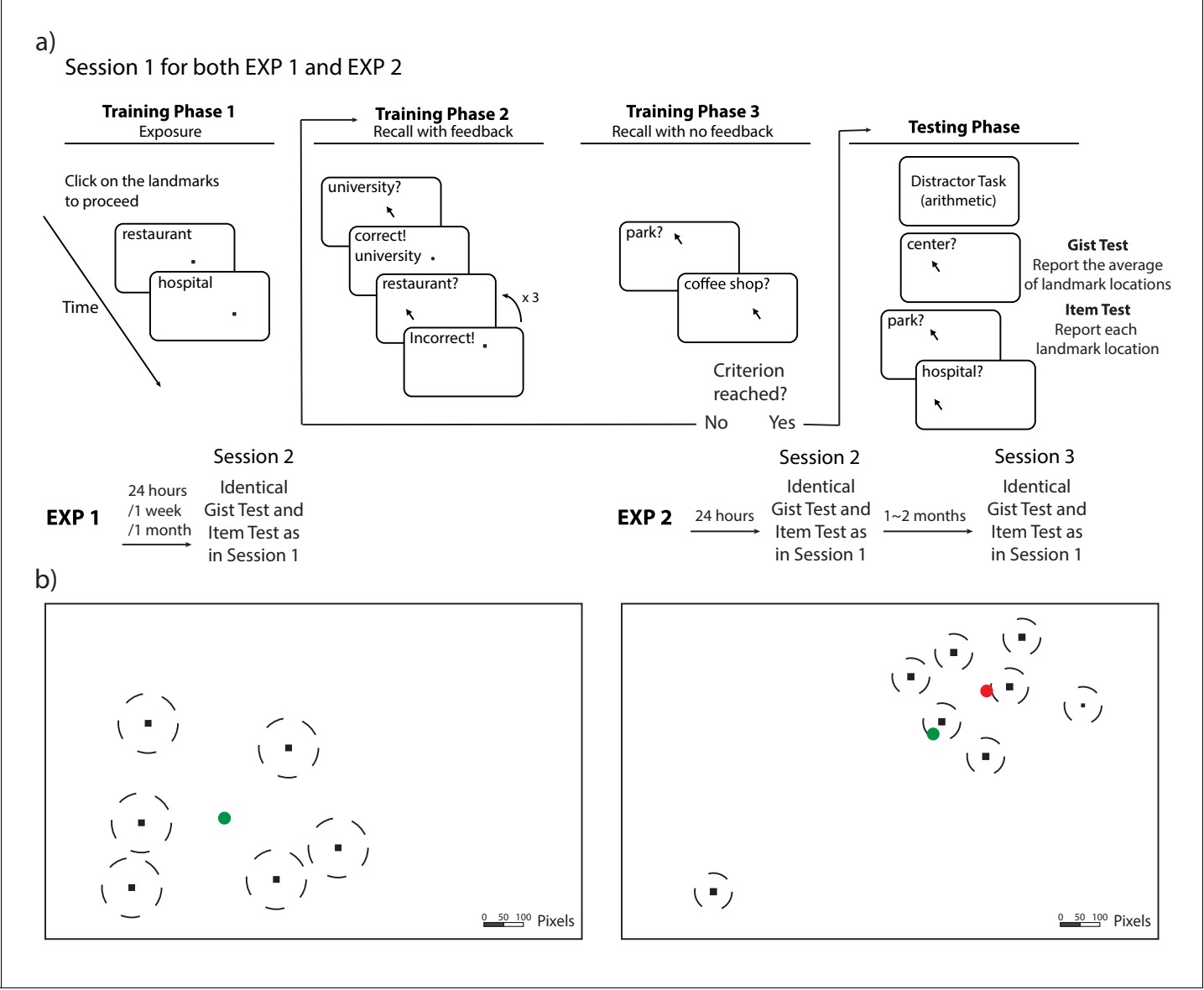

**Figure 1.** Procedure and stimuli for Experiments 1 and 2. (**a**) Schematic illustration of the procedure for Experiments 1 and 2. The procedure of Session 1 is the same for Experiments 1 and 2 (with the exception of the number of trials). Participants completed cycles of encoding (with feedback) and evaluation (without feedback) until they could retrieve each landmark individually within the training criteria. (**b**) An illustration of the location of the stimuli (drawn to scale) for Experiments 1 and 2. The locations (black dots) were the same for all participants, but the mapping between the location and landmark name was randomized for each participant. The dash lines around the dots indicate the training criteria (80 pixels for Experiment 1 and 50 pixels for Experiment 2). The green circle indicates the center of these encoded locations and the red circle indicates the 'local' center of the encoded locations (excluding the outlier) in Experiment 2.

memory is biased toward the gist representation (*Figure 3a*). We compared the observed bias computed from data collected from participants at each time point with an *item-only* simulated bias, which assumed learners only had item memory, such that the magnitude of error for each simulated retrieved location would be the same as the corresponding item memory collected from participants, but the direction of the simulated location would be random. We also compared the observed bias to an item-plus-gist simulated bias which assumed both item memory and an additional influence from the gist memory (abbreviated as *gist simulated bias* below for simplicity), such that each simulated retrieved location would be generated from the same error as in the item-only simulation but the probability of a retrieved location being simulated is weighted by its distance toward the reported center. Therefore, a location that is closer to the center will have higher

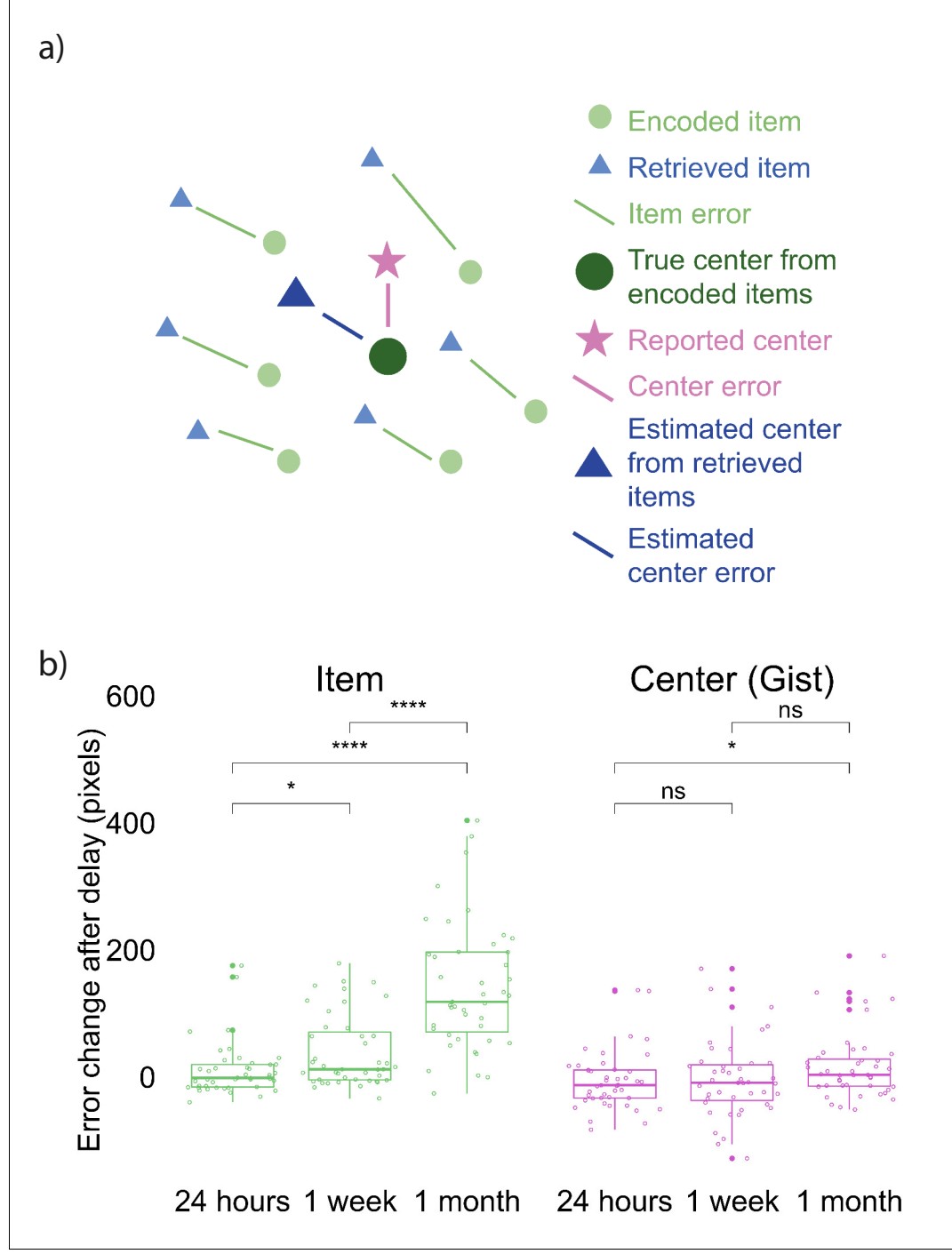

**Figure 2.** Error measurements and results. (**a**) Error measurements. (**b**) Change in error by group and memory type (the band indicates the median, the box indicates the first and third quartiles, the whiskers indicate ± 1.5 × interquartile range, and the solid points indicate outliers). Greater values indicate an increase in error in Session 2 over Session 1. * indicates p < 0.05 and **** p < 0.0001 by Mann-Whitney tests. *Figure 2—figure supplement 1* shows the absolute error for both item and gist memory at Sessions 1 and 2. *Figure 2—figure supplement 2* shows the error change over time in reported gist, estimated gist, and simulated gist based on a simple item-only simulation (discussed at the end of Experiment 1 result section).

The online version of this article includes the following source data and figure supplement(s) for figure 2:

**Source data 1.** Experiment 1 error change over time.

**Figure supplement 1.** Experiment 1 error in item and gist memory at Session 1 (**a**) and at Session 2 (**b**).

*Figure 2 continued on next page*

*Figure 2 continued*

**Figure supplement 1—source data 1.** Experiment 1 Session 1 error.
**Figure supplement 1—source data 2.** Experiment 1 Session 2 error.
**Figure supplement 2.** Experiment 1 error change between Session 1 and 2 in gist memory, estimated gist, and simulated estimated gist based on a simple item-only simulation.
**Figure supplement 2—source data 1.** Experiment 1 error change in reported gist, estimated gist, and simulated estimated gist from an item-only simulation.

probability of being retrieved in the simulation (see Materials and methods for details). We used the reported center (rather than the true center of encoded items) in this analysis because we found a decrease in accuracy of the reported center after 1 month compared to 24 hr, as discussed above. In the next section 'Follow-up bias analyses', we repeated the bias analysis using the true center of encoded items, for consistency with common practices in ensemble perception research (*Brady and Alvarez, 2011*; *Lew and Vul, 2015*).

In order to examine whether participants' item memory became more dissimilar over time to what would be expected from having item representations and no separate gist representation, we conducted a 2 (session: Session 1 vs. Session 2) x 3 (delay groups: 24 hr, 1 week, and 1 month) ANOVA for the difference between the observed bias and item-only simulated bias. This test revealed a significant interaction between delay group and session, $F(2, 254) = 3.53$, $p = 0.03$, indicating that the difference between item-only simulated bias and the observed bias significantly increased over time (a, b, c, d). Follow-up t-tests for all delay groups revealed that for the 1-month group only, the difference between the observed bias and the item-only simulated data was significantly higher for Session 2 compared to Session 1, $t(80.76) = 3.18$, $p < 0.01$. No across-session comparisons for any other delay groups were significant (ps > 0.65). Furthermore, only the observed bias at 1 month was significantly greater than the item-only simulated bias, $t(42) = 3.73$, $p < 0.001$ (Fig 3b).

In order to examine whether participants' item memory became more similar over time to what would be expected from having both item and gist representations, we conducted an analogous 2 x 3 ANOVA for the difference between gist simulated bias and the observed bias. This test revealed a significant interaction between delay group and session, $F(2, 254) = 6.33$, $p < 0.01$, indicating that the difference between gist simulated bias and the observed bias significantly decreased over time. Follow-up t-tests for all delay groups revealed that for the 1-month group only, the difference between the observed bias and the gist simulated bias was significantly lower for Session 2 compared to Session 1, $t(82.50) = 4.39$, $p < 0.001$. No across-session comparisons for any other delay groups were significant (ps > 0.65). (Note. Because the level of gist influence added to the gist simulated bias was arbitrarily selected, we did not expect the endpoint value for the observed bias to match the gist simulated bias; we will return to this point in the Discussion). These results indicate that participants' biases were increasingly consistent with the assumption of a separate gist representation and increasingly inconsistent with reliance only on item memories. By 1 month (but not after 1 day or 1 week), item memory retrieval was biased toward the reported gist [We used this bias analysis to measure the influence of the gist—as opposed to comparing the distance between the reported center and estimated center or their error difference over time—for two reasons. First, our measure of bias shows the magnitude of the particular direction the items were attracted to. Second, the estimated center was calculated from an aggregation of item memories, which may already have been influenced by the center of these items at a delay (as shown by the bias analysis). Therefore, comparing the estimated center and the reported center over time would not allow us to isolate the influence of the center. To demonstrate this point, we computed an estimated center error from the item-only simulation. Such simulated estimated center error is significantly different from participants' reported center error and estimated center error, which suggests that the estimated center computed from the item memories was influenced by the center (*Figure 2—figure supplement 2*)].

Taken together, the results of Experiment 1 reveal that although there is a persistent relation between item and gist memory during memory consolidation, the nature of this relation changes over time. We suggest that early in memory consolidation, retrieval of gist depends on the successful retrieval of individual items, but then, as item memory weakens over time, the relatively stronger

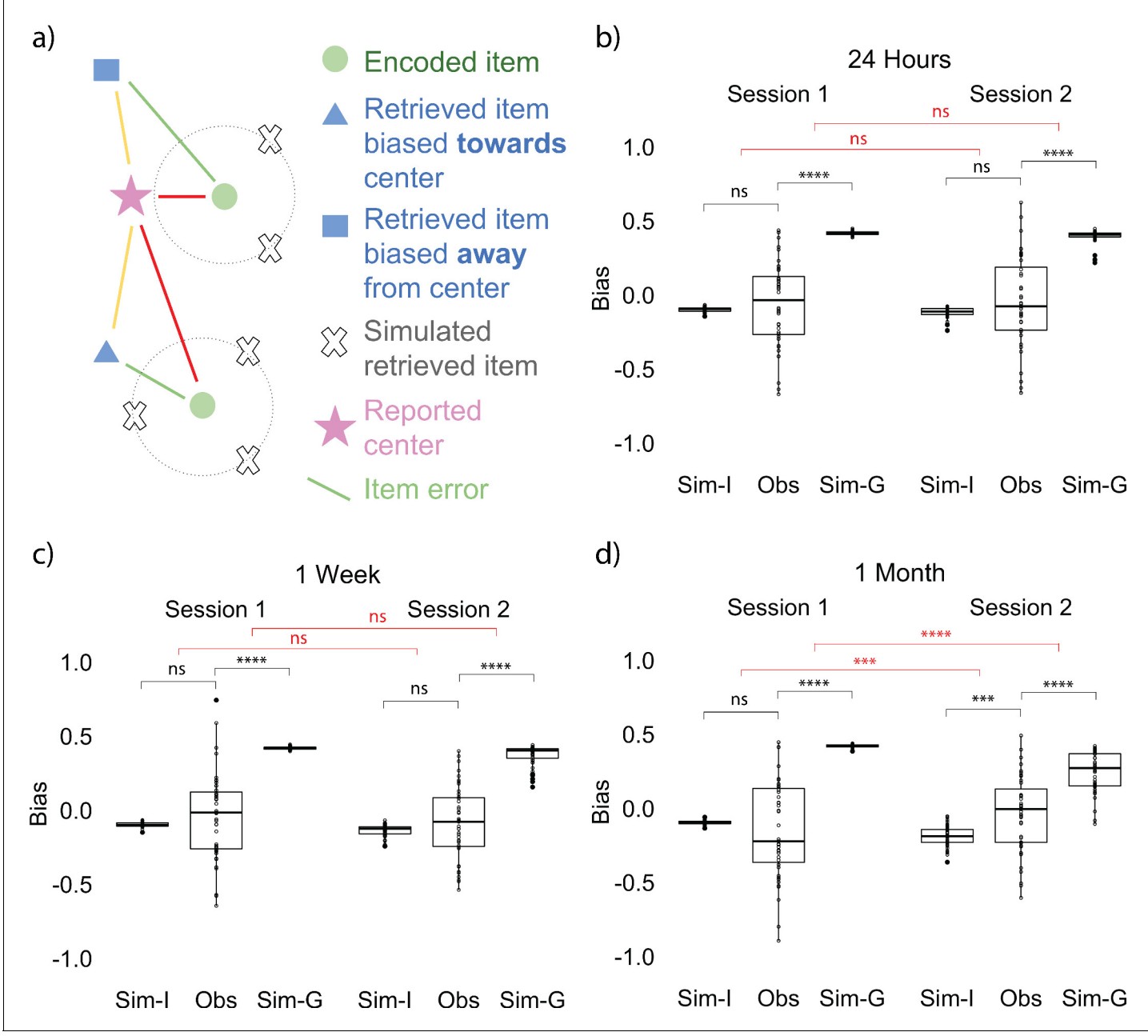

**Figure 3.** Bias measurement and results at each session. (a) Bias measurement. The bias for each recalled location is (red - yellow) / green. The blue square is an example of a recalled item that is biased away from the reported center and the blue triangle is an example of a recalled item that is biased toward the reported center. Bias for each participant is an average of bias for all the locations. (b, c, d) Item-only simulated bias (Sim-I), observed bias (Obs), and gist simulated bias (Sim-G) at each session for delay groups of 24 hr, 1 week, and 1 month (the band indicates the median, the box indicates the first and third quartiles, the whiskers indicate ± 1.5 × interquartile range, and the solid points indicate outliers). * indicates p < 0.05, ** p < 0.01, *** p < 0.001, and **** p < 0.0001 by t-tests between observed bias and simulated biases (black) and t-tests comparing the difference in observed bias and simulated biases between sessions (red).

The online version of this article includes the following source data for figure 3:

**Source data 1.** Experiment 1 bias at each session.

gist memory begins to guide retrieval of item memory. As a consequence, this new gist representation can exert influence over memories in ways described by reconstructive memory theories.

## Follow-up bias analyses

In order to test factors that may influence the gist-based bias and its generalizability, we conducted the bias analyses as discussed above with the three following modifications (see Materials and methods for additional details): First, to be consistent with common practices in ensemble perception research (*Brady and Alvarez, 2011*; *Lew and Vul, 2015*), we repeated the bias analysis using the center of encoded items (big green circle in *Figure 2a*) instead of the reported center. Second, the assumption that each item is weighted the same may be overly simplistic and the weight of the items may influence the representation of the gist and the bias results. Therefore, we computed a center that was a weighted average based on the accuracy of each item, such that items with higher recalled accuracy would be weighted more in the computed center compared to items with lower accuracy. We repeated the bias analysis using this weighted center. Third, the mental representation of the locations that participants encoded may not be a linear transformation of the actual item locations on the computer screen, and this nonlinearity may account for the observed biases in location memory. To capture the potential non-linear warping of the stimulus space, we generalized the Euclidean error measure to a Minkowski's measure, where error $d(a, b) = \sqrt{[1.5](a_1 - b_1)^{1.5} + (a_2 - b_2)^{1.5}}$, and conducted the same bias analysis.

Across the three analyses, we found the same pattern: The observed bias became more and more dissimilar from the item-only simulated bias, indicated by a significant interaction between delay group and session in the three ANOVAs (all $F$s > 6.42, ps < 0.01), and became more and more similar to the gist simulated bias over time, indicated by a significant interaction between delay group and session in ANOVAs (all $F$s > 6.81, ps < 0.01). Furthermore, the observed bias at 1 month differed from the item-only simulated bias (all $t$s > 3.85, ps <0.001), but not at other delays. In addition, the bias computed under these three approaches are highly correlated with the bias with reported center in the prior section (all $r$s > 0.9, ps < 0.001), suggesting the varied approaches generated similar bias to the bias in the prior section. In summary, the result of increasing gist-based bias over time replicates in analyses using the center of encoded locations, weighted center based on item accuracy, and with a Minkowski's measure in non-Euclidean space.

## Experiment 2

The stimuli and procedure of Experiment 2 (*Figure 1*) were similar to those of Experiment 1 but differed in two major ways (see Materials and methods for more details). Firstly, we used a repeated measures design in Experiment 2 so that we could observe changes in memory at short (1 day) and long (1–2 months) retention intervals within each subject (N = 43). Secondly, one of the landmarks was an 'outlier', meaning that its location fell far out of the range of the cluster where the majority of the landmarks were (see Experiment 2 item locations in *Figure 1*). The inclusion of an outlier location enabled us to examine the influence that a single 'atypical' item has not only on the initial estimation of the center (as in *Haberman and Whitney, 2010*; *Richards et al., 2014*; *Whitney and Yamanashi Leib, 2018*) but also on the source of bias in item memory at long delays. In addition, in Session 1, item memory was derived from the item memory test in the last round of evaluation during training (see the procedure for Experiment 2 in *Figure 1*) to streamline the session.

## Gist memory decreased less than item memory over time

In Experiment 2, we used the same error measurement for the accuracy of item and gist memory as in Experiment 1 (*Figure 2a*; see Materials and methods for details). Participants performed above chance in both item (i.e. the average of distance between encoded item locations and center of the screen; the average of distance between encoded item locations and center of encoded item locations) and gist memory (i.e. the distance between the center of encoded locations and center of the screen) tests at all sessions (all p < 0.0001; *Figure 4—figure supplement 1*). To examine how item and gist memory changed over time, we conducted a 2 (delay: short retention of 24 hr, long retention of 1–2 months) x 2 (memory type: item, center) aligned ranks transformation ANOVA with repeated measures of error change. This test revealed a main effect of delay, $F(1, 126) = 16.20$, p < 0.001, memory type, $F(1, 126) = 68.96$, p < 0.001, and an interaction between delay and memory

type, $F(1, 126) = 27.40$, p < 0.001. This interaction indicates that the error of individual item retrieval increased more over time compared to the reported center (*Figure 4a*). Specifically, whereas item memory error change was higher after 1-2 months compared to 24 hr by Wilcoxon signed rank test ($Z = 5.28$, p < 0.0001), no such significant difference was detected for gist memory error change ($Z = 0.72$, p = 0.47). Experiment 2 thus replicated the finding observed in Experiment 1 that item memories decreased significantly more relative to gist memory over time.

## Positive relationship between item and gist memories across time

To explore the relation between item and gist memory, we fit a linear mixed effects model on reported center error with fixed effects of delay (24 hr and 1–2 months), estimated center error, and their interaction with a random effect of participant to account for repeated measures within participants. We found a significant effect of estimated center error ($SSE = 29114.3$, $F(1, 69.92) = 12.08$, p < 0.001), but not a main effect of delay ($SSE = 1420.9$, $F(1, 59.59) = 0.59$, p = 0.45) or an interaction between the estimated center error and delay ($SSE = 1102.9$, $F(1, 68.80) = 0.46$, p = 0.50). The result of a persistent relationship between estimated center error and reported center error at short and long retention intervals was replicated in a within-participants design.

## Item memory retrieval was biased toward the local gist over time

To examine whether the influence of gist on item memories changes over time, we applied the same bias analysis as in Experiment 1, using participants' reported center of all the retrieved items as bias center (*Figure 1b*). We conducted a one-way repeated measures ANOVA on the difference between the observed bias and item-only simulated bias across sessions (after training, 24 hr, and 1 month)

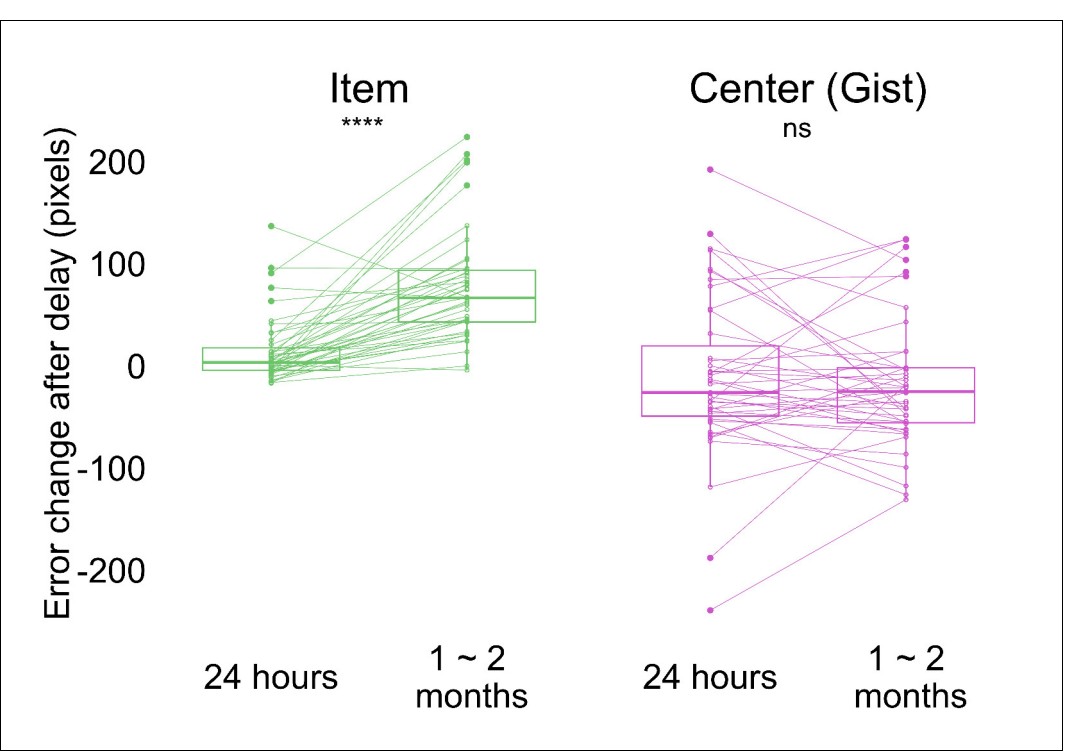

**Figure 4.** Change in error by delay and memory type (the band indicates the median, the box indicates the first and third quartiles, the whiskers indicate ± 1.5 × interquartile range, and the solid points indicate outliers). Greater values indicate an increase in error from Session 1 after delay. **** p <0.0001 by Wilcoxon signed rank tests. Dots and lines indicate participants. *Figure 4—figure supplement 1* shows the absolute error for both item and memory at all sessions.

The online version of this article includes the following source data and figure supplement(s) for figure 4:

**Source data 1.** Experiment 2 error change over time.

**Figure supplement 1.** Experiment 2 error in item and gist (center) memory at each session.

and also the same ANOVA on the difference between the observed bias and gist simulated bias. In contrast to Experiment 1, the difference between item-only simulated bias and the observed bias did not significantly change over time, $F(2, 84) = 1.34$, p = 0.28 (*Figure 5a*). Furthermore, unlike in Experiment 1, the observed bias was not significantly higher compared to the item-only simulated bias at long retention, $t(42) = -0.47$, p = 0.64 (*Figure 5a*). At the same time, the difference between gist simulated bias and the observed bias only marginally decreased over time, $F(2, 84) = 3.04$, p = 0.05 (*Figure 5a*). This may be driven by an unpredicted negative bias (i.e. bias away from the reported gist) immediately after learning, $t(42) = -2.09$, $p = 0.04$, and after a short retention interval, $t(42) = -2.27$, p = 0.03, revealed by t-tests against the item-only simulation.

What might explain the different bias results between Experiments 1 and 2? We suspect this is a result of the outlier item. Prior work in visual working memory research showed that outliers were discounted in estimating the gist (*Haberman and Whitney, 2010*). In our Experiment 1, where there was not an outlier, we saw that the item retrieval was biased toward the center of all of the items; however, in Experiment 2, the center of *most* of the items would be the local center excluding the outlier (*Figure 1b*). It is possible that for participants in Experiment 2, the items were biased toward the local clustering center excluding the outlier.

In order to test this possibility, we conducted an analysis that computed the gist-based bias of the items using the local center (the true center from the encoded items disregarding the outlier). As in Experiment 1, the difference between item-only simulated bias and observed bias significantly increased over time, $F(2, 84) = 8.51$, p < 0.001 (*Figure 5b*). Follow-up paired t-tests showed that the difference between item-only simulated bias and observed bias at long retention was significantly higher compared to the difference after training, $t(42) = -3.67$, p < 0.001, and compared to the difference at short retention, $t(42) = -2.87$, p < 0.01. The same comparison between after training and 24 hr was not significant, $t(42) = -1.49$, p = 0.14.

On the other hand, the difference between gist simulated bias and observed bias decreased over time, $F(2, 84) = 13.80$, p < 0.001 (*Figure 5b*). Follow-up paired t-tests showed that the difference between gist simulated bias and observed bias at long retention was significantly smaller than the difference after training, $t(42) = -4.59$, p < 0.001, and compared to the difference at short retention, $t(42) = -3.81$, p < 0.001. The same comparison between after training and 24 hr was not significant, $t(42) = -1.62$, p = 0.11. Furthermore, only the observed bias at the long retention interval was significantly greater than the item-only simulated bias, $t(42) = 2.28$, p = 0.03 (*Figure 5b*). This increased bias was observed even for the outlier item: Retrieval of the location of the outlier item was significantly more biased toward the local center after a long retention interval compared to a short retention interval, revealed by a comparison to the item-only simulated bias ($Z = 2.46$, p = 0.01). These results indicate that item memories in Experiment 2 were biased, at long retention intervals, toward the center as in Experiment 1, but that the 'center' in Experiment 2 was not the global center but instead the local center excluding the outlier item.

## Over-weighting of the outlier in gist memory

Our analysis of item bias suggests that the outlier is 'discarded' as a member of the cluster of locations, which is consistent with some prior studies (e.g. *Ariely and Carmon, 2003*; *Haberman and Whitney, 2010*); however, other work has shown that outliers can greatly disrupt or shift the representation of a set of events (*Richards et al., 2014*). Could both be happening in this paradigm? In order to explore the influence of the outlier on the representation of gist, we applied a weighted model adapted from working memory literature and computed an estimation of the weight of the outlier in the reported gist (*Haberman and Whitney, 2010*; see Materials and methods for details).

The estimated weight of the outlier was significantly higher than 0.125 (i.e. the level assuming equal weights across all items) immediately after learning, $t(42) = 2.14$, p = 0.04, after a short retention interval, $t(42) = 2.89$, p < 0.01, and after a long retention interval, $t(42) = 3.83$, p < 0.001, (*Figure 6*). The change in outlier weight after short compared to long retention intervals did not significantly differ, $t(42) = 0.92$, p = 0.36. In other words, the outlier has not been discarded from the set, but quite to the contrary, the outlier had a disproportionate influence on the explicit retrieval of gist after a delay. In contrast, the implicit effect of the center on bias in item retrieval seems to emerge from a center that is uninfluenced by the outlier (*Figure 5b*). These results revealed that the outlier consistently influenced participants' reported center more than other items at all tested time points.

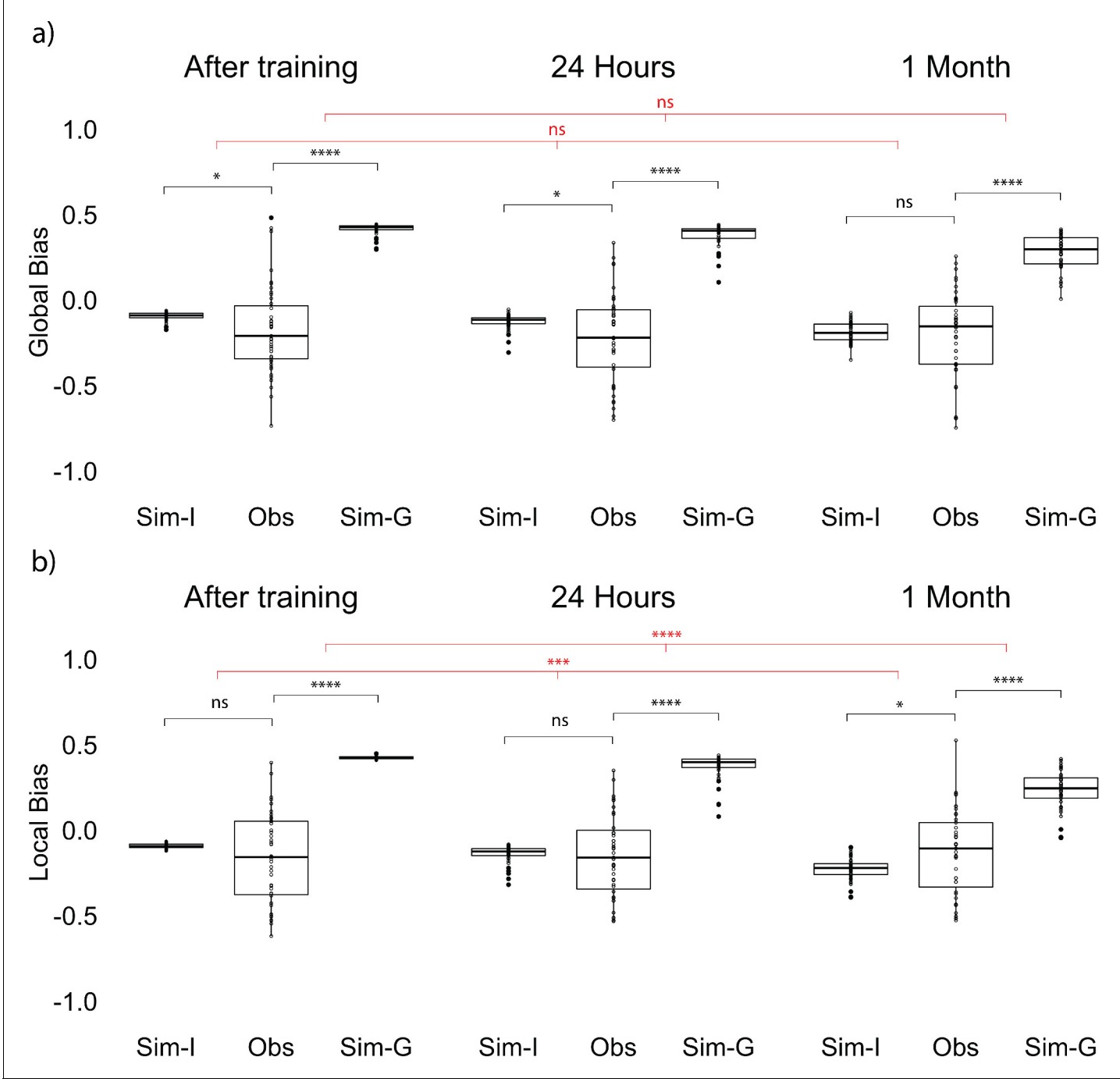

**Figure 5.** Global and local bias at each session. (a) Global observed bias (Obs), item-only simulated bias (Sim-I), and gist simulated bias (Sim-G) at each session. Global bias uses the report center. (b) Local observed bias (Obs), item-only simulated bias (Sim-I), and gist simulated bias (Sim-G) at each session. Local bias excludes the outlier item when estimating the center. The band indicates the median, the box indicates the first and third quartiles, the whiskers indicate ± 1.5 × interquartile range, and the solid points indicate outliers. * indicates $p < 0.05$, *** $p < 0.001$, and **** $p < 0.0001$ by paired t-tests between observed data and simulation (black) and repeated measures ANOVA comparing the difference in data and simulations across sessions (red).

The online version of this article includes the following source data for figure 5:

**Source data 1.** Global bias at each session.

**Source data 2.** Local bias at each session.

Taken together, Experiment 2 replicated the main findings from Experiment 1 that gist memory decreased less compared to item memories over time (*Figure 4*) and a positive relationship between item and gist memories in a within-subject design. The 'outlier' item changed the relationship between items and reported global center after a long retention interval. By 1–2 months, items were no longer biased toward the global reported center which overweighted the outlier. Instead, they were biased toward the local center excluding the outlier over time.

## Discussion

We examined how human learners extract the 'gist' (generalities, common properties, or summary statistics) across individual instances, and how memory for these instances and for the gist evolve and influence each other over time through two behavioral experiments spanning one to two months. We demonstrated that the accuracy of item memory (memory for spatial locations on a screen) decreased more compared to the accuracy of gist memory (center of the locations) over time, although there was a persistent positive correlation between them. Critically, item memories were increasingly biased toward the gist over time. Participants' biases grew less similar over time to a simulation relying only on memory for individual items and more similar to a simulation assuming a separate gist representation. In the presence of an outlier item, the local gist, excluding the outlier, became the source of bias, instead of the gist participants directly reported, which consistently over-weighted the outlier across time. We think that gist memories, initially built from item memory, gradually developed to guide item memory as their relative strength changed over time.

Consistent with prior research (*Antony and Stiver, 2020*; *Berens et al., 2020*; *Lutz et al., 2017*; *Posner and Keele, 1970*), item memory became less accurate over time while gist memory remained relatively intact over time. Our findings converge with this prior research even when using an explicit instruction to retrieve gist memory, rather than inferring gist from another measure as in prior research. A shortcoming of prior research is that the relation between item and gist memory over time is rarely assessed. We showed that the relationship between item memory and gist memory persisted across delay periods despite decreased accuracy in item memory. This relationship could have resulted from the influence of gist memory on the retrieval of item memory, from the influence of item memory on gist memory, or both. Our gist-based bias results shed light on the direction of this relationship: Item memory retrieval was biased toward the gist only after 1 month, which suggested that the correlation at 1 month was likely to be due to the influence of gist memory on the retrieval of item memory.

Our findings that items are increasingly biased toward the gist as the accuracy of item memories decreases over time provide new evidence for the memory reconstruction framework, which proposes that memory retrieval is a combination of different sources with varying strength (*Brady et al., 2015*; *Hemmer and Steyvers, 2009*; *Huttenlocher et al., 2000*; *Tompary et al., 2020*). Our work extends prior evidence of increased schematization in memory consolidation (*Richards et al., 2014*; *Richter et al., 2019*; *Tompary et al., 2020*) by demonstrating a new form of influence from the gist on item memory: gist-based bias. In contrast to prior memory consolidation research that showed increased schematization earlier than 1 month (*Graves et al., 2020*; *Richter et al., 2019*; *Tompary et al., 2020*), the gist-based bias in our current work did not increase by 24 hr or 1 week. This discrepancy could be because the intensive training participants experienced in our paradigm increased the strength of item memories relative to gist memory during learning, and only after a long retention interval did the strength of item memory decrease to an extent that allowed bias to manifest.

The increased bias may reflect a slow systems consolidation process that results in a qualitatively different memory representation after longer retention intervals (*McClelland et al., 1995*; *Richards et al., 2014*). An increased reliance on neocortical areas over time would be expected to strengthen gist memory, to the extent that neocortex tends to represent information in a 'semanticized' form (*Sekeres et al., 2017*). The results are also consistent, however, with a change in reliance on different forms of memory within the same memory systems. The current results are not diagnostic on this point — they are consistent with a range of theories on the interplay between episodic and semantic memories over time (*Renoult et al., 2019*; *Richards et al., 2014*; *Robin and Moscovitch, 2017*; *Winocur and Moscovitch, 2011*; *Sekeres et al., 2018*).

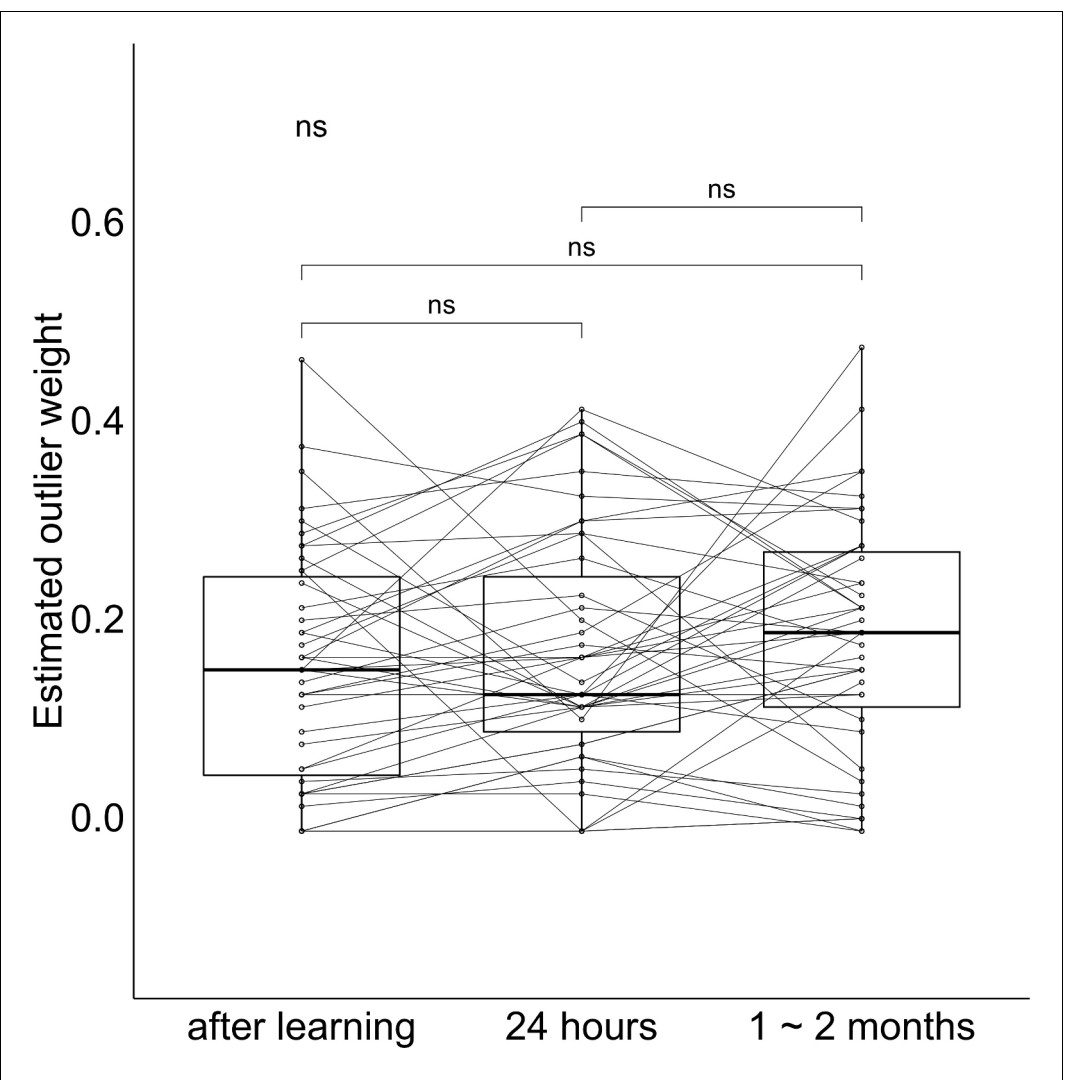

**Figure 6.** Outlier weight values at each session. The band indicates the median, the box indicates the first and third quartiles, the whiskers indicate ± 1.5 × interquartile range. Dots and lines indicate participants.
The online version of this article includes the following source data for figure 6:

**Source data 1.** Outlier weight at each session.

Our results of increasing gist-based bias over time parallel visual working memory work, which shows evidence of a hierarchical organization of memory: items are more biased toward their center as uncertainty increases in order to increase the overall precision of retrieval (*Brady and Alvarez, 2011*; *Lew and Vul, 2015*; *Orhan and Jacobs, 2013*). Our results detected a similar gist-based bias in long-term memory consolidation. Moreover, in Experiment 2, after a long retention interval, the reported gist overweighted the outlier, whereas the item memories were biased toward the local gist which discounted the outlier. This finding also mirrors prior ensemble perception results that outliers are discounted or excluded in estimating summary statistics (*de Gardelle and Summerfield, 2011*) and suggests that the gist influencing item retrieval is not a simple average of the items. The results might reveal two different sampling strategies for gist extraction. Because participants had explicit knowledge about the outlier, they might have given more weight to the outlier in explicitly recalling and reporting the gist, similar to the change in the pattern by inconsistent items observed in long-term memory work (*Richards et al., 2014*). In contrast, the local center that influenced the items might reflect an implicit representation with a sampling strategy discounting the outlier, consistent with findings in perception work (*de Gardelle and Summerfield, 2011*; *Haberman and*

*Whitney, 2010*). Our results suggest that visual working memory and long-term memory might be underpinned by a similar reconstructive mechanism and open up new directions to bridge the two fields.

One limitation of the current experiments is that the testing order (i.e. gist memory before item memory) might have encouraged the retrieval of the items to be consistent with the gist (*Tversky and Kahneman, 2019*; *Mutluturk and Boduroglu, 2014*). We initially chose this order because we were most interested in the change in gist representation and wanted to minimize the influence of item memories on gist estimation in later recall. We also were concerned that the extent that these two tests influenced the other was not symmetric; in other words, the influence of item memory on gist might be more pronounced than the influence of gist on item memory. Because the testing order is the same for the three different delay intervals, we reasoned that the changes in item memory and bias across delay groups could not simply be a result of the order. In addition, in Experiment 2, the items were not biased toward the center participants reported, suggesting that even if the gist test occurring before item test influences the recall for the items, the influence may be minor. However, multiplicative effects, such as floor or ceiling effects present only at one time point, could still influence the results and the influence from the testing order may still exist. Although the testing order is likely not to influence the change over time, the bias for all delay groups may be lower overall under the reverse testing order. More studies with counterbalanced testing order will be helpful to evaluate this possibility.

Future research can be done to test the generality of our findings to other domains of human cognition. It would be interesting to explore whether our findings, which considered gist memory as a spatial average, would generalize to a broader definition of gist, such as gist-like memory for events (*Moscovitch et al., 2016*). For example, when first learning what a birthday party is from attending a few, the 'gist' representation of a birthday party may be dominated by memory for a few parties, but over time the gist becomes a more stable representation that can influence retrieval of those specific birthday party events. In addition, the dissociation of gist-based bias in Experiment 2 also mirrors the dissociable implicit and explicit attitude in social categories (*Gawronski and Bodenhausen, 2006*). More work could further disentangle these processes in long-term memory consolidation, which could enlighten our understanding of the cognitive mechanism underlying the formation of gist in social categories.

We began by posing the question of how one extracts a summary statistic from individual instances, but without a doubt, the summary statistic we have used here to answer this question—the arithmetic average of x,y coordinates—is overly simplistic. Firstly, the item-only simulations in our work are oversimplified compared to the item-only models in the categorization literature (*Nosofsky, 1988*). There surely could be other more sophisticated item-only models that can fit our data. However, our results put a new constraint (i.e. a gist-based bias) for item-only models in long-term memory. Secondly, our implementation of a gist representation in our bias simulations was very simplistic. For example, we assumed an arbitrary amount of gist influence, implementing a more qualitative than quantitative assessment of the presence of a gist representation. Additional experiments with more within-subject statistical power could be used to constrain models that quantify the precise amount of gist influence (as a parameter in individual model fits). Thirdly, this gist influence may be influenced by many other factors, such as the variability in item locations, the accuracy and the confidence of item memories, the distance from the items to the screen center or to the boundary (*Intraub and Richardson, 1989*), individual differences in cognitive functions (e.g. executive control), and the demand characteristics of explicitly recalling the center. The current design did not allow for enough variability to tease apart these possibilities, but future research systematically manipulating these factors will be helpful in addressing these issues. Finally, and perhaps most importantly, the principles that govern aggregation of individual spatial locations do not in any obvious way translate to the nature of summary statistics for other episodic memories (like that average bear!). Although there is much work to be done to understand the ways in which we aggregate information across multiple experiences, the current experiments should provide a useful launching off pad for future explorations of this question.

In summary, we have shown that memory for individual items and memory for the gist of a set of items changed over the course of long-term memory consolidation. We propose that the gist that was initially extracted from item memories gradually started to guide item memory retrieval over longer durations as their relative memory strength changed. These findings bridge research in areas

of cognitive science ranging from perception and working memory to episodic and semantic memory, providing important new insights into our ability both to learn about distinct events and to generalize across similar experiences.

## Materials and methods

### Experiment 1

#### Participants

In Experiment 1, we recruited 147 members of the University of Pennsylvania community (18–30 years old; normal or corrected to normal vision) to participate in the experiment for monetary compensation. Participants selected to sign up for a second session that followed their first session by either 24 hr, 1 week, or 1 month [Because participants were not randomly assigned into three different delay conditions, a difference in expectation may influence their learning and consolidation. We did not find evidence, however, for any differences in behavior between groups at initial learning (*Figure 2—figure supplement 1*). Also, the results from Experiment 1 replicated in Experiment 2 with a within-subject design]. Sample size was based on Experiment 2 which was conducted first [Namely, we set the number of subjects after exclusion in Experiment 2 as a minimum sample size. After we reached the sample size, we continued to recruit participants until the end of the academic term]. We excluded 10 participants because of low performance on Session 1 (i.e. reported gist was out of the scope of the learned landmarks) and then seven participants because of individual and gist performance of any sessions lower than 3 *SD* below average. Our reported results thus include 130 participants, with 44 participants in the 24 hr group (age: *M* = 21.3, *SD* = 2.9, gender: 61% females), 43 participants in the 1-week group (age: *M* = 21.9, *SD* = 2.6, gender: 67% females), and 43 participants in the 1-month group (age: *M* = 21.4, *SD* = 2.0, gender: 74% females). All procedures were approved by University of Pennsylvania IRB (IRB #705915, Linguistic and Nonlinguistic Functions of Frontal Cortex).

#### Procedure

The experimental procedure is displayed in *Figure 1*. All participants completed Sessions 1 and 2; the only difference between groups was the time delay between sessions. Session 1 included training and testing. During training, participants were trained to retrieve six landmark locations consecutively on a laptop until their retrieval error for each landmark was fewer than 80 pixels in any direction. Eighty pixels was chosen to be the criterion because it was less than 1/2 of the shortest distance between any pairs of the encoded locations, and thus would ensure that participants could differentiate the locations in recall. The training included three phases. In Phase 1, the landmarks appeared on the screen one at a time, and participants were required to click on each landmark to proceed. *Figure 1* illustrates the landmark locations; note, on each trial, only one location was presented (never the full map), and the center of the encoded locations was never presented to participants. In Phase 2, we asked participants to recall the location for each landmark by clicking on the screen when given its name as a cue, and we gave them feedback about their guesses: participants had three attempts to retrieve each landmark location. For each attempt, if the distance between the encoded location and retrieved location satisfied the training criterion (i.e. 80 pixels), the correct location would be shown on the screen; otherwise, a message would be prompted that their attempt was incorrect. The correct location would be shown on the screen after three incorrect attempts. In Phase 3, participants recalled each landmark consecutively without feedback, one at a time. If each of the retrieved landmarks fell in the range of 80 pixels, the participant could proceed to testing; if not, the participant was redirected back to Phase two to receive more training. After participants reached the training criterion, they completed 10 unrelated arithmetic problems, in order to minimize potential influences from working memory. Finally, participants were tested on their memory of the locations: They indicated their guess about the center of the landmarks (gist memory test), with an instruction 'Indicate the center (average location) of the landmarks you have seen'. Then, they separately recalled each landmark location (item memory test, which was identical to the recall procedure in Phase 3). The order of items was randomized in all phases in training and testing. In both tests, participants were incentivized to be accurate through bonus payments. They would receive a bonus of 1 dollar for the gist memory test if their error was within 100 pixels and a

bonus of 1 dollar for the item memory test if their average error across all items was within 80 pixels. All trials were self-paced. The total time for Session 1 was approximately 12 min.

After 24 hr, 1 week, or 1 month, participants returned for Session 2. Session 2 was identical to the gist test and item test in Session 1, in which participants first reported the center and then the location of each landmark. Trials were again self-paced. The total time for Session 2 was approximately 5 min. Participants could choose to quit the experiment after Session 1 and receive 10 dollars for their time, otherwise they would be paid after Session 2. The payment ranged from 16 to 20 dollars, depending on participants' performance in their gist memory and item memory test.

## Error measurement

In order to measure the accuracy for item memory (memory for each landmark), gist memory (reported memory for the center of the landmarks), and the estimated gist (center of all the retrieved items), we developed three error measurements as follows.

Item Memory Error (green line in *Figure 2a*): The error for each item was defined as the Euclidean distance between the retrieved location for each landmark and its encoded location, where $d$ (a, b) $= \sqrt{(a_1 - b_1)^2 + (a_2 - b_2)^2}$. Each participant's item memory error was computed as the average error for the six landmarks. We defined two chance performance. Chance performance based on the center of the screen was 348 pixels, which was determined by the average Euclidean distance between the center of the laptop screen and each encoded item location. This distance corresponded to what participants' performance would be if they only remembered the center of the screen and just clicked the center of screen when asked to recall an item. Mathematically, this distance corresponded to the average error a participant would have if they guessed anywhere on screen. Chance performance based on the center of the encoded items was 267 pixels, which was determined by the average Euclidean distance between the center of encoded item locations and each encoded item location. This distance corresponded to what participants' performance would be if they only remembered the center of item locations and just clicked that center when asked to recall an item.

Gist Memory Error (purple line in *Figure 2a*): The error for gist memory was defined as the reported center error, which was the Euclidean distance between the participant's reported center and the true center of all the encoded items. Chance performance was 270 pixels, which was the Euclidean distance between the center of the laptop screen and the true center of all encoded locations. This distance corresponded to what participants' performance would be if they just clicked the center of screen when asked to report the center.

Estimated Gist Memory Error (blue line in *Figure 2a*): The error for the estimated gist based on items was defined as the estimated center error, which was the Euclidean distance between the center of each participant's retrieved item locations and the true center of all encoded locations. In other words, the estimated center can be thought of as what the participant's gist estimate would be if it were directly computed by averaging across all retrieved item locations.

## Bias measurement

In order to measure the influence of gist on item memory, we developed a bias measurement as follows. Since the error analysis revealed a decrease in gist memory (i.e. reported center) after a month, there could be a difference in using reported center and using the true center of encoded items as bias center. We initially used the reported center as the center for the bias analysis. However, we also computed a bias using the center of encoded items to be consistent with common practices in ensemble perception research (*Brady and Alvarez, 2011*; *Lew and Vul, 2015*).

## Observed bias

The bias toward the center for each retrieved item was defined as the relative difference in distance between a participant's reported center and each landmark's encoded location versus each landmark's retrieved location. This relative difference was then divided by the error for that landmark:

$$observed\,bias = \frac{(d(\mathrm{Encoded\,Item, Reported\,Gist}) - d(\mathrm{Retrieved\,Item, Reported\,Gist}))}{d(\mathrm{Encoded\,Item, Retrieved\,Item})}$$

where $d$ (a, b) $= \sqrt{(a_1 - b_1)^2 + (a_2 - b_2)^2}$ (*Figure 3b*). Bias thus can range between $-1$ and 1 and bias

> 0 indicates that item memory is biased towards the center while bias < 0 indicates that the item is biased away from the center. Each participant's observed bias was computed as the average across the biases of the six landmarks.

## Item-only simulation

The item-only simulation assumed that the magnitude of error for each item memory would be the same as the corresponding item memory collected from participants, but the direction of simulated recalled location would not be systematically influenced by the gist. Following this assumption, we generated 1000 simulations for each participant. Each simulation consisted of six simulated retrieved items, corresponding to all the six landmark locations. For each location, we randomly generated a retrieved location based on the participant's true error for this specific location, allowing its angle relative to the encoded location to vary randomly across the simulations (*Figure 7b*; gray cross). If a simulated location fell outside the boundaries of the screen, the algorithm generated a new location. The bias value for each of the 1000 simulations was the average across each simulation's six retrieved locations. The item-only simulated bias for each participant was the average across the 1000 simulations.

Item-plus-gist simulation (abbreviated as gist simulation for simplicity): The gist simulation assumed that the magnitude of error for each item memory would be the same as the corresponding item memory collected from participants, but the reported center would systematically influence the probability of recalled locations (instead of the uniform probability distribution around the item location in the 'item-only simulation'). Following this assumption, we generated 1000 simulations for each participant. Each simulation consisted of six simulated retrieved items, corresponding to all the six encoded landmark locations. For each encoded location, the simulated retrieved location was generated based on not only the participant's true error for this specific location, but also based on a probability assigned according to the distance from that simulated location to the reported center as follows (*Figure 7a*). For any encoded location, the space where a simulated retrieved location can possibly be generated would be a circle centering the encoded location with a radius of an error from participants' error data. We divided the circle into 200 angles. At each angle on that circle, we calculated the distance from that angle to the reported center and assigned a probability to that angle based on such distance:

$$P_i = \frac{d(simulated\ item_i, reported\ center) - \max(d(simulated\ item_i, reported\ center))}{\sum\limits_{i=1}^{n}(d(simulated\ item_i, reported\ center) - \max(d(simulated\ item_i, reported\ center)))}$$

where *i* corresponds to each angle and *n* corresponds to the total number of angles (200). The probability for any location to be retrieved would thus be inverse to the distance between the angle and the reported center. If a simulated location fell outside the boundaries of the screen, the algorithm would generate a new location. The bias value for each of the 1000 simulations was the average across each simulation's six retrieved locations. The gist simulated bias for each participant was the average across the 1000 simulations.

For both item-only simulations and gist simulations, when item memory error increases, the increased error will lead to a negative bias value despite no meaningful bias away from the center of the landmarks (*Figure 7c*). Therefore, it is necessary to compare the data with the simulations.

## Statistics

To examine whether gist memory persisted when memory for items decayed over time, we conducted a 3 (group: 24 hr, 1 week, and 1 month) X 2 (memory type: item, center) aligned ranks transformation ANOVA, a nonparametric approach that allows for analyzing main effects and interaction (*Kay and Wobbrock, 2020*), of the error change (Session 2 error values - Session 1ne error values) and also two-tailed Mann-Whitney tests for between-group error change comparisons, because the data were not normally distributed as determined by a Shapiro-Wilk test. In order to examine whether there was a relation between item and gist memory, we used a linear model to evaluate the effects of estimated center error, delay group, and their interaction on reported center error. To interpret the effects of the overall effect of the delay group on the gist error, rather than the individual effects of each group, we reported the SSE rather than betas.

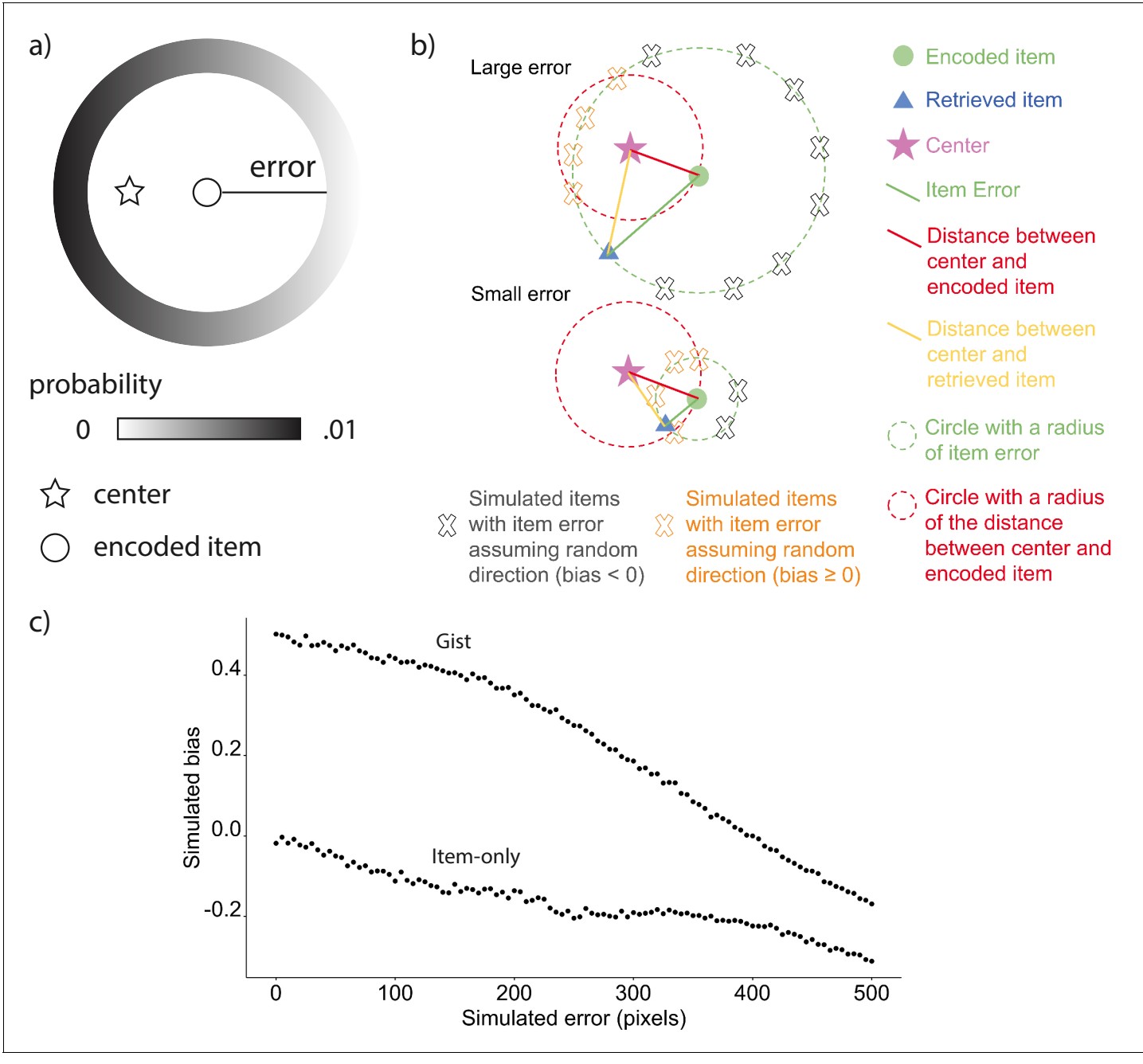

**Figure 7.** Simulated bias controlling for error. (**a**) An example of the probability of simulated locations to be generated for an encoded location given the same extent of error in gist simulation. (**b**) Items with large errors are more likely to have a negative absolute bias by chance. First, the proportion of the arc with negative absolute bias in the circumference of simulated items is higher for items with large error. Because the distance between any points on the arc defined by the intersection between the green circle and red circle to the center will be shorter than the red distance, the points will all have an absolute bias value ≥ 0 (indicated by orange X marks), whereas the points outside of the arc will have a negative bias (indicated by gray X marks). Second, even though the retrieved item (blue triangle in the lower figure) with small error and the retrieved item with large error (blue triangle in the upper figure) are biased in the same direction, the absolute bias for the retrieved item with the small error is positive, whereas the other is negative, which demonstrates how a retrieved item with large error could cause negative bias without meaningfully being biased away from the center relative to its encoded location. (**c**) Simulations based on the six encoded locations in Experiment 1 showed that random error of retrievals not assuming direction was negatively correlated with bias for both item-only simulation and gist simulation.

The online version of this article includes the following source data for figure 7:

**Source data 1.** The relation between simulated error and bias.

In order to examine whether the observed bias became more dissimilar to bias predicted by the item-only simulation over time, we conducted a 2 (session: Session 1 vs. Session 2) x 3 (delay groups: 24 hr, 1 week, and 1 month) ANOVA in the difference between the observed bias and item-only simulated bias (observed bias - item-only simulated data). As follow-up analyses, we used two-tailed paired t-tests to compare the difference in observed bias and item-only simulated bias between session 1 and session 2 for each delay group. In order to examine whether item retrievals were biased toward the center at any time points, we compared the observed bias against the item-only simulated bias for each group at each session.

In order to examine whether the observed bias became more similar to bias predicted by the gist simulated bias, we conducted the same ANOVA in the difference between the observed bias and gist simulated bias (gist simulated data - observed bias). As follow-up analyses, we used two-tailed paired t-tests to compare the difference in observed bias and gist simulated bias between session 1 and session 2 for each delay group. Note that due to the limited number of trials, we could not fit the most accurate parameter of gist influence and therefore the amount of gist influence under the gist simulation is arbitrary and likely not accurately reflecting the amount of gist influence in observed data. Therefore, unlike the analysis for item-only simulated bias, we did not predict that over time the observed bias would become indistinguishable from the gist simulated bias and we did not test whether observed bias and gist simulated bias significantly differed at all sessions for the delay groups. Reports were not corrected for multiple comparisons.

### Follow-up bias analyses

Center of Encoded Locations as Bias Center: This analysis was exactly the same with the bias analysis in the prior section, except that this analysis used the center of encoded locations, instead of the center participants reported as bias center.

Weighted Center by Item Accuracy: This analysis was exactly the same with the bias analysis in the prior section, except that this analysis used a center weighted by item accuracy as bias center, instead of the center participants reported as bias center. The weight of each item was determined by (1- error of the item/error of all items)/(number of items - 1), such that items with higher accuracy would be weighted more in the computed center and also that the weight of all items summed up to 1.

Minkowski's Space: This analysis was exactly the same with the bias analysis in the prior section, except that all the distance was computed by $d(a,b) = \sqrt[1.5]{(a_1 - b_1)^{1.5} + (a_2 - b_2)^{1.5}}$. We selected 1.5 because Minkowski distance is defined by $\sqrt[g]{(a_1 - b_1)^g + (a_2 - b_2)^g}$ and g is typically selected between 1 and 2.

### Swapped Items

In recalling the location for the landmarks, participants might 'misbind' the label of a landmark and its location (e.g. indicate the location of the 'restaurant' at the actual location for the 'university' and vice versa). In order to test the potential influence of such errors on our results, we developed a criterion to identify pairs of items that were swapped, and we swapped them back to see if that changed the results. That is, for example, if (1) the retrieval for 'restaurant' was closest to the encoded location for 'university', (2) the retrieval for 'university' was closest to the encoded location for 'restaurant', (3) the retrievals were both within the range of both of the encoded locations (i.e. the distance between encoded 'restaurant' and 'university' / 2) and, (4) there were no other retrievals in this range, we then swapped the retrieved university and restaurant responses and used the swapped results for the analyses described above. We found that swapping the items did not change any of the reported results.

## Experiment 2

### Participants and procedure

We recruited 77 members of the University of Pennsylvania community (18–30 years old; normal or corrected to normal vision) to participate in the experiment for monetary compensation. Sample size was based on prior behavioral memory studies [Because this was a new experiment, we were unable to identify an effect size from a past study that was appropriate for a power analysis. Therefore, we tried to collect a sample equivalent to what is commonly collected in behavioral memory studies

(e.g. *Schapiro et al., 2017*) and continued to recruit participants across two semesters]. All procedures were approved by University of Pennsylvania IRB (IRB #705915, Linguistic and Nonlinguistic Functions of Frontal Cortex). All 77 participants received training and testing during Session 1 and reported item and center memories again after 24 hr (Session 2). Sessions 1 and 2 were identical with Experiment 1, except that in Experiment 2 during Session 1, participants were trained to retrieve eight landmark locations, one of which was a spatial outlier (see Experiment 2 stimuli in *Figure 1*), until their retrieval error for each landmark was fewer than 50 pixels (again, a distance less than ½ of the shortest distance between the pairs of encoded locations) in any direction. In addition, in Session 1 of Experiment 2, to streamline the session, item memory was derived from the item memory test in the last round of evaluation during training (Phase 3), which was immediately followed by the gist memory test (see Experiment 2 procedure in *Figure 1*). The time for Session 1 was approximately 25 min, which was longer than that for Experiment 1 because in Experiment 2, participants learned more locations and the training criterion was harder (50 pixels, as opposed to 80 pixels in Experiment 1).

After 32–57 days, 50 participants returned for Session 3 by email invitation. Session 3 was identical to Session 2 (i.e. participants reported their memory for the center and then each item). The time for Session 2 and 3 was approximately 10 min. Of the 50 participants who returned for the third session, one participant was excluded because their individual and gist performance for at least one session was lower than 3 *SD* below average. We did not exclude participants whose reported gist memory error was larger than the distance between the screen center and the true center at Session 1, as in Experiment 1, because in Experiment 2, a large gist error could be a meaningful result that reflects the overweighting of the outlier in reporting gist. We excluded six participants who placed the outlier where the majority of items were, which means the error of the outlier was larger than 573 pixels (i.e. the distance between the center of screen and outlier encoded location). The reason we excluded these participants was that in Experiment 2, if participants swapped the outlier with one of the other items, or simply put the outlier among the other items, this outlier swap would strongly inflate the bias value toward the global reported center, which does not necessarily reflect a true bias toward the center. Our reported results thus include 43 participants (age: *M* = 21.5, *SD* = 2.2, gender: 75% female).

### Error measurement

All error measures were calculated as in Experiment 1, except that the chance performance for individual items based on the center of the screen was 386 pixels (determined by the average Euclidean distance between the center of the laptop screen and each encoded item location), the chance performance for items based on center of encoded locations was 262 pixels (determined by the average Euclidean distance between the center of the encoded item locations and each encoded item location), and the chance performance for gist memory was 223 pixels (determined by the average Euclidean distance between the center of the laptop screen and the center of all encoded item locations).

### Bias measurement

We calculated bias as in Experiment 1, except that we additionally computed a local gist bias, which was a bias index using the local center (i.e. the center of the seven encoded locations excluding the outlier) as the bias center.

### Outlier weight estimation

In order to estimate the weight of the outlier on the reported gist memory, we developed a weighted model as follows: For each participant at each session, a series of weights were applied to the encoded outlier location. The range of the weight of the outlier was from 0 to 1, with a stepwise increment of 0.0125. The weight for each of the other encoded items was assumed to be the same and would thus be (1 - outlier weight)/number of items that were not the outlier, ranging from 0 to 0.125. Based on these weights, 81 simulated centers were computed: when the outlier weight was 0, the simulated center would be a perfect local center ignoring the outlier; when the outlier weight was 0.125, the simulated center would be a perfect global center of all items, since there were eight items in total; when the outlier weight was 1, the simulated center would be the outlier itself.

For each participant at each session, the Euclidean distance between each simulated center and reported center was computed, resulting in 81 distances. We used the weight that resulted in the smallest distance as the estimated weight of the outlier for that participant at that session.

## Statistics

As in Experiment 1, we conducted a 2 (delay: short retention of 1 day or long retention of 1–2 months) x 2 (memory type: item, center) aligned ranks transformation ANOVA with repeated measures of error change (short, defined by Session 2 error values - Session 1 error values, or long, defined by Session 3 error values - Session 1 error values) and pairwise Wilcoxon signed rank tests for error comparisons, since change in error was not normal as determined by a Shapiro-Wilk test.

As in Experiment 1, in order to examine whether there was a relation between item and gist memory, we used a linear mixed effects model on reported center error with fixed effects of delay (24 hr and 1–2 months), estimated center error, and their interaction, as well as a random effect of participant.

As in Experiment 1, in order to examine whether the item retrievals were increasingly biased toward the reported center over time, we conducted a one-way repeated measures ANOVA in the difference between observed bias and item-only simulated bias across sessions (after training, 24 hr, and 1 month). We compared the observed bias against the item-only simulated bias at each session by paired t-tests to examine whether item retrievals were significantly biased toward the reported center. We conducted the analogous ANOVA analysis in the difference between the observed bias and gist simulated bias using the reported center as bias center. As explained in the results, we then conducted the same statistical analyses using the local center as the bias center and then follow-up paired t-tests between sessions. Reports were not multiple comparisons corrected.

In order to examine whether the outlier was weighted more, the same, or less compared to other items, we compared the outlier weight values against the weight assuming all items to be equal (i.e. ⅛ = 0.125) with t-tests. In order to examine whether the weight of the outlier in gist memory changed over time, we used a paired t-test comparing the outlier weight change after a short retention interval (Session 2 outlier weight values - Session 1 outlier weight values) against the outlier weight change after a long retention interval (Session 3 outlier weight values - Session 1 outlier weight values).

## Acknowledgements

The authors gratefully acknowledge the data collected by Siqi Lin, Jennifer Nazario, Emily Potter and the helpful discussions with colleagues in Penn Psychology. This work was supported by National Institute of Health (5R01DC009209-14, 5R01DC015359-05) awards to Sharon Thompson-Schill.

## Additional information

### Competing interests

Anna C Schapiro: Reviewing editor, *eLife*. The other authors declare that no competing interests exist.

### Funding

| Funder | Grant reference number | Author |
| --- | --- | --- |
| National Institutes of Health | R01 DC009209-14 | Sharon L Thompson-Schill |
| National Institutes of Health | R01 DC015359-05 | Sharon L Thompson-Schill |

The funders had no role in study design, data collection and interpretation, or the decision to submit the work for publication.

## Author contributions
Tima Zeng, Conceptualization, Data curation, Software, Formal analysis, Investigation, Visualization, Methodology, Writing - original draft, Project administration, Writing - review and editing; Alexa Tompary, Conceptualization, Formal analysis, Methodology, Writing - review and editing; Anna C Schapiro, Conceptualization, Methodology, Writing - review and editing; Sharon L Thompson-Schill, Conceptualization, Resources, Supervision, Funding acquisition, Methodology, Writing - review and editing

## Author ORCIDs
Tima Zeng  https://orcid.org/0000-0003-3805-4701
Alexa Tompary  https://orcid.org/0000-0001-7735-3849
Anna C Schapiro  https://orcid.org/0000-0001-8086-0331
Sharon L Thompson-Schill  https://orcid.org/0000-0002-9750-4306

## Ethics
Human subjects: That informed consent, and consent to publish, was obtained. The specific ethical approval obtained from University of Pennsylvania IRB (IRB #705915, Linguistic and Nonlinguistic Functions of Frontal Cortex). The guidelines were followed. The above information was described in the Materials and Methods.

## Decision letter and Author response
Decision letter https://doi.org/10.7554/eLife.65588.sa1
Author response https://doi.org/10.7554/eLife.65588.sa2

# Additional files
## Supplementary files
• Transparent reporting form

## Data availability
All data generated or analysed during this study are included in the manuscript and supporting files.

The following dataset was generated:

| Author(s) | Year | Dataset title | Dataset URL | Database and Identifier |
|---|---|---|---|---|
| Zeng T | 2020 | Tracking the relation between gist and item memory over the course of long-term memory consolidation | https://osf.io/jxme8/ | Open Science Framework, jxme8 |

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
