## [Decision Letter]

**Acceptance summary:**

This paper addresses an important and as yet unresolved question in the cognitive science of memory, namely how do we extract gist or summary representations across related experiences, and how does this relationship change over time? Using innovative behavioural paradigms, the authors demonstrate that gist memory remains robust after 1 month, while item memory becomes increasingly biased towards the gist representation. This provides important new clues as to how we consolidate memories of experiences and their gist.

**Decision letter after peer review:**

Thank you for submitting your article "Tracking the relation between gist and item memory over the course of long-term memory consolidation" for consideration by *eLife*. Your article has been reviewed by 3 peer reviewers, one of whom is a member of our Board of Reviewing Editors, and the evaluation has been overseen by Chris Baker as the Senior Editor. The following individuals involved in review of your submission have agreed to reveal their identity: Karsten Rauss (Reviewer #2); Richard Henson (Reviewer #3).

*Reviewer #1:*

Zeng and colleagues present an interesting and timely exposition of how memory for items and generalities across related experiences are formed and influence each other. Across two carefully designed longitudinal studies spanning 1-2 months, the authors integrate insights from ensemble perception research to develop a novel landmark learning paradigm in which a set of landmarks are clustered together. Participants were required to learn the specific location of each landmark (item memory) as well as the spatial centre of the locations (reported gist). Leveraging hierarchical clustering models, the authors computed a gist-based bias measurement, enabling them to comment on the extent to which gist memory influences memory for specific items, as well as an index of estimated centre assembled from the retrieval of individual items. This enabled the authors to establish the amount of gist information available in item memories, and to tease apart the direction of the relationship between item and gist memory. I particularly appreciated Study 2's exploration of how the presence of an "outlier" item in spatial location impacts the gist and the relationship between item and gist representations from Study 1. This innovative approach allowed them to determine the extent to which the gist is robust against outlier items over time.

Overall, I enjoyed reading this manuscript. It elegantly addresses an important and as yet unresolved question in cognitive science, namely the extent to which gist representations become independent of individual item memories as they are extracted during encoding and how this relationship potentially changes over time. The inclusion of a longitudinal dimension further enables us to understand something of the temporality of consolidation over short versus longer time periods. The manuscript is very well written, the experimental studies have been meticulously conducted using a novel interdisciplinary approach, limitations are appropriately acknowledged, and the conclusions drawn are highly appropriate and measured. I believe this study will make a very nice contribution to the memory literature and will certainly spur new lines of enquiry in this field.

*Reviewer #2:*

It is often said that the amount of information humans can commit to long-term memory is essentially unlimited. However, this large storage capacity partly rests on our ability to compress new information before and during memory consolidation. One way to achieve such compression is by storing the common denominator of related experiences, which is often referred to as gist memory. During later recall, the brain is thought to deduce individual memories from the stored gist representation, rather than reading them out one by one. This not only reduces the memory footprint associated with learning, it also supports adaptive generalization of learning to novel situations.

Zeng and colleagues address two important questions regarding gist memory: First, how do gist memory traces develop over time in comparison to memory traces for individual items? And second, how do the two classes of memory traces influence each other?

The authors developed an experimental protocol in which healthy participants learn the association between landmark labels (e.g. university, park, restaurant) and abstract locations on a computer screen. Memory is subsequently measured in terms of the precision with which participants are able to indicate the learned locations, as well as their geometric center (i.e., the gist location).

A major strength of this approach is that it allows for precise, quantitative assessments of item and gist memory, and how they influence each other. On the other hand, the abstract nature of the task makes it difficult to generalize the results to other forms of gist memory. Moreover, gist memory is explicitly assessed directly after initial learning, which does not allow for clear conclusions as to whether similar memory traces would have developed spontaneously. Nevertheless, the protocol represents a valuable tool for studying the interaction of related memory traces over time.

The authors performed two experiments using this task. In the first experiment, they show that gist memory is more stable over one month compared to item memory; and that gist and item memory are positively related across all measured time-points (i.e., after 24 hours, one week, and one month). However, the nature of this positive correlation changes over time, with gist memory biasing item memory only after one month. The results of a second experiment corroborate these conclusions. They additionally indicate that outlier items (i.e., landmarks that are spatially distant from the cluster of all other landmarks) affect explicit, but not implicit gist memory, with only the latter biasing memories for individual items. Taken together, these results corroborate the notion that the importance of gist memories in guiding recall increases over time. Importantly, the data also provide an estimate of the relevant time window, i.e. between one week and one month after learning.

My only major concern is that the experimental task provides a rather restricted view of gist memory. Thus, it remains unclear what the results mean for other kinds of gist memory, both visual and otherwise. The authors discuss this point on p.16 and offer some interesting speculations. However, the main problem I see is that memory for gist locations may reflect a combination of basic perceptual strategies (as acknowledged by the authors in their references to ensemble perception) and the demand characteristics of the task. This issue is most prominently seen in the difference between the effects for global and local centers observed in Experiment 2. Here, the instruction to explicitly recall the center is what may have created (or at least emphasized) the memory for the global center in the first place. More generally, the fact that gist memory was tested first and early on after learning may have changed the associated memory trace, making it difficult to generalize these findings to cases in which gist memories emerge spontaneously. While this may be the price to pay for precise, quantitative error and bias measures of errors, the authors should discuss these limitations in more detail.

*Reviewer #3:*

A fundamental puzzle in human memory has been whether we retain individual experiences, and extract their gist by pooling over those experiences only when the gist is needed, or whether we also store a separate representation of the gist (regularities). This study uses locations of landmarks on a screen to argue that such a gist representation (the central tendency of the 2D locations of landmarks) is extracted gradually over time during memory consolidation, i.e., over a period between one day and a month after initial encoding of the landmark locations. Furthermore, the study suggests that this gist representation is not influenced by atypical locations, suggesting it is more than the simple average of all item memories, and that atypical stimuli may be encoded and/or retrieved separately.

I think it would really improve the paper if the authors could compare a number of simple computational models, and show the model corresponding to their favoured conclusion (i.e., that a separate gist representation increases its influence on memory over time) is the only one that is consistent with their data. Ideally, this would entail quantitative fitting of parametrised models, but failing that, it might be sufficient to demonstrate qualitatively that certain models can never explain critical results (like the bias measure above, or the effect of an outlier). For example, I presume that an item-only model in which random noise (in both x,y directions) increased with delay could explain their first result of a greater effect of delay on reported item error (Ir) than Gr? But is this model consistent with their regression results, and most importantly, could it never reproduce their bias results?

Another concern is whether the same results would hold if participants' representation of gist (Gr) were more than the simple average of their reported item locations (Ge). For example, would there be any consequences for the authors' conclusions if some 2D locations were represented more accurately than others – e.g., one might expect the location of landmarks close to the edge of the screen to be stored more accurately than ones closer to the centre of the screen (which could be tested by whether edge locations need fewer training trials?). I appreciate that locations are trained to the same criterion, but it is nonetheless possible that some representations are still more precisely encoded than others even after such criterion-training. Then if participants' reported centre were the weighted average of the reported item locations, weighted by each location's remembered precision, could that affect the authors' bias measure? I appreciate that this would not explain why the difference between reported and true centre changes over retention interval, but if one allowed location and precision information to decay at different rates, could this cause such an interaction with retention interval? This could be another model to simulate?

It is unfortunate that the authors did not counterbalance the order of item and gist memory tests. They do consider this limitation in the Discussion, and note that any additive effect of test order would not explain the interaction between memory type and delay, but of course the presence of nonlinear/multiplicative effects (e.g., floor/ceiling effects) means it is not sufficient to conclude that their results could not depend on test order, i.e., it is always better to empirically test generalisation by running the other counterbalancing. This is particularly important here, where there are good a priori, theoretical reasons why test order might matter, e.g., shift people from strategies based on retrieving items or retrieving gist, depending on what type of memory is probed first. So while the authors state that this could be tested in future studies, the paper would be stronger if the authors could run this counterbalancing themselves and show the same results, so that future researchers do not waste time trying to replicate effects that turn out to be conditional on test order.

Essential revisions:

1. As the authors note, a central debate in many fields of cognitive psychology is whether gist is extracted and stored separately from items, or whether only items are stored, and gist is estimated on-the-fly by retrieving multiple items. For example, can memory performance be explained by an exemplar-only model (like Nosofsky's), or is a separate prototype representation needed? I thought that this paradigm offered a neat way to address this debate by comparing the gist reported by participants with the gist estimated from all their item reports. While the two might differ due to random noise, if this difference systematically increased with the retention interval, this would seem to support a separate representation of gist (prototype), whose influence increases with delay/consolidation. However, the authors do not appear to test this simple effect of delay on the difference score between reported gist error (Gr) and estimated gist error (Ge). Is there a reason they didn't do this? They do regress Gr against Ge and Delay, but I'm not sure why regression is better to test the above hypothesis. In any case, the positive relationship between Gr and Ge, but lack of a significant interaction between Ge and Delay, would seem more compatible with an exemplar-only model, i.e., no need for a separate representation of gist. This is where the authors' bias measure may be critical, but I could not intuit whether their finding of increased bias with delay towards Gr (relative to true gist, Gt) rules out an exemplar-only model.

2. I think it would really improve the paper if the authors could compare a number of simple computational models, and show the model corresponding to their favoured conclusion (i.e., that a separate gist representation increases its influence on memory over time) is the only one that is consistent with their data. Ideally, this would entail quantitative fitting of parametrised models, but failing that, it might be sufficient to demonstrate qualitatively that certain models can never explain critical results (like the bias measure above, or the effect of an outlier). For example, I presume that an item-only model in which random noise (in both x,y directions) increased with delay could explain their first result of a greater effect of delay on reported item error (Ir) than Gr? But is this model consistent with their regression results, and most importantly, could it never reproduce their bias results?

3. Another concern is whether the same results would hold if participants' representation of gist (Gr) were more than the simple average of their reported item locations (Ge). For example, would there be any consequences for the authors' conclusions if some 2D locations were represented more accurately than others – e.g., one might expect the location of landmarks close to the edge of the screen to be stored more accurately than ones closer to the centre of the screen (which could be tested by whether edge locations need fewer training trials?). I appreciate that locations are trained to the same criterion, but it is nonetheless possible that some representations are still more precisely encoded than others even after such criterion-training. Then if participants' reported centre were the weighted average of the reported item locations, weighted by each location's remembered precision, could that affect the authors' bias measure? I appreciate that this would not explain why the difference between reported and true centre changes over retention interval, but if one allowed location and precision information to decay at different rates, could this cause such an interaction with retention interval? This could be another model to simulate?

4. A related concern is the more general problem that the mental representations of item locations may not be a simple linear transform of the true locations. If the mental space of locations is a nonlinearly warped version of the physical space (even if still diffeomorphic, e.g., expanded towards the edges of the physical space, rendering edge items more distinct), then not only will the reported item locations (Ir) be systematically biased with respect to their true locations, but the simple average of the reported item locations would not match the reported gist location. Take the simple case of a 1D space with two items encoded at location x=1 and x=3, such that the gist is x=2, but presume the mental representation of those locations were warped by x^2^: then the reported item locations would be 1 and 9 (before downscaling to the original min/max of the space), resulting in an estimated gist (Ge) of 5, whereas the gist in participants' memory (as reported in Gr) would be 2^2^ = 4. I think this potential nonlinear warping of the stimulus space can be captured to some extent by generalising the Euclidean error measure used by the authors to Minkowski's measure (for 2D vectors) = ((x1-x2)^g^ + (y1-y2)^g^)^1/g^ (where g=gamma=2 for Euclidean, but could be fit to allow non-Euclidean distances). See for example Tversky and Gati, Psych Review (1982) for further discussion about differences between physical and psychological spaces. When implemented in an item-only model (and the presence of noise that increases with delay), could such nonlinearities/non-Euclidean measures allow such a model to fit the results, particularly the authors' bias measure? Perhaps not, or perhaps the authors can explain such considerations are irrelevant to their results, but I think some thought and ideally modelling about this possibility might be worthwhile.

5. The results from the outlier in Experiment 2 lead the authors to say themselves that "the gist influencing item retrieval is not a simple average of the items" – so they clearly need a more complex model in which outliers are down-weighted according to the type of memory question asked, and I wasn't 100% clear what that model was, i.e how exactly the authors explain all the results of both Experiments. I think this is where some simple computational models extended to handle outliers, would be very helpful, in terms of mechanistic understanding.

6. Regarding the authors' critical bias measure, they write it on page 20 as:

"(Encoded Item – Reported Gist) – (Retrieved Item – Reported Gist) / (Encoded Item – Retrieved Item)". Firstly, parentheses around the numerator are missing, i.e. it should be: ((Encoded Item – Reported Gist) – (Retrieved Item – Reported Gist)) / (Encoded Item – Retrieved Item) at least, compared to the code in their bias_func.m, and compared to what would be necessary for the bias to range between 0 and 1. Secondly, if evaluated literally, this measure would always be 1, ((It – Gr) – (Ir – Gr)) = (It – Ir), which is identical to the denominator (where It = true or encoded item locations). I realise this "equation" was meant to give a conceptual idea rather than literal definition, but I think it would be safer to give the precise definition, which is ( d(It,Gr) – d(Ir,Gr) ) / d(It,Ir) where d = some distance function (and It, Ir and Gr are 2D vectors), and in the authors case, they use a Euclidean distance (as in Point 4 above), i.e. d (a,b)=(a1−b1)2+(a2−b2)2.

7. More importantly, please could the authors help me understand the rationale for this specific bias measure, and why it is so critical for establishing the presence of a gist representation? I appreciate that Gr can be replaced with Ge, but why is this combined measure of item memory better than the simple Gr – Ge gist measure in terms of distinguishing theories? Apologies if I have missed something fundamental to the paper, in which case, perhaps explain a bit more in the Introduction why this particular bias measure is so critical?

8. It is unfortunate that the authors did not counterbalance the order of item and gist memory tests. They do consider this limitation in the Discussion and note that any additive effect of test order would not explain the interaction between memory type and delay, but of course the presence of nonlinear/multiplicative effects (e.g., floor/ceiling effects) means it is not sufficient to conclude that their results could not depend on test order, i.e., it is always better to empirically test generalisation by running the other counterbalancing. This is particularly important here, where there are good a priori, theoretical reasons why test order might matter, e.g., shift people from strategies based on retrieving items or retrieving gist, depending on what type of memory is probed first.

9. The experimental task provides a somewhat restricted view of gist memory. Thus, it remains unclear what the results mean for other kinds of gist memory, both visual and otherwise. While the authors discuss this point on p.16, memory for gist locations may reflect a combination of basic perceptual strategies (as acknowledged by the authors in their references to ensemble perception) and the demand characteristics of the task. This issue is most prominently seen in the difference between the effects for global and local centres observed in Experiment 2. Here, the instruction to explicitly recall the centre is what may have created (or at least emphasized) the memory for the global centre in the first place. The fact that gist memory was tested first and early on after learning may have changed the associated memory trace, making it difficult to generalize these findings to cases in which gist memories emerge spontaneously. While this may be the price to pay for precise, quantitative error and bias measures of errors, the authors should discuss these limitations in more detail.

10. The authors touch on the subject of neural substrates in the Discussion however, I would like to have seen slightly more consideration of this topic. For example, I was surprised by the statement on page: 15 that "neocortex tends to represent information in a 'semanticized' form" – this strikes me as somewhat at odds with more recent theoretical positions on the interplay between episodic and semantic memory (e.g., Renoult et al. TICS 2019) as well the processes by which memory transformation from item to gist memory occurs (Robin and Moscovitch, 2017). While not the focus of this manuscript, it would be helpful for the authors to consider these contemporary positions in relation to their findings.

11. As a point of curiosity, have the authors considered the potential role of individual differences in some of these processes and whether some individuals may be more adept at extracting gist versus item representations?

12. Is there any relationship of the present results to Brainerd and Reyna's "Fuzzy Trace Theory" (https://en.wikipedia.org/wiki/Fuzzy-trace_theory), which is based on the role of verbatim and gist representations in memory?

[Editors' note: further revisions were suggested prior to acceptance, as described below.]

Thank you for submitting your article "Tracking the relation between gist and item memory over the course of long-term memory consolidation" for consideration by *eLife*. Your article has been re-reviewed by a Reviewing Editor and Chris Baker as the Senior Editor.

Essential revisions:

There was some concern regarding the extent to which some of the comments raised by the reviewers had been addressed in the previous revision. The below issues have been raised by the reviewers as warranting serious consideration.

(i) The failure to counterbalance the order of gist and item tests was not felt to have been satisfactorily discussed in the Discussion. The reviewers all agree that the paradigm is interesting and relevant, but the lack of counterbalanced replication based on an explicit power analysis limits the generalisability of these findings. The authors need to temper their claims and more deeply acknowledge and address the limitations of this study.

(ii) More emphasis needs to be placed on the modelling, moving the models to the main text and discussing these more fully. The reviewers had suggested simulating a range of models, yet only the simplest of these models was simulated (exemplar model with noise), and this was placed in Supplementary Material. While a weighted version was explored and an alternative Minkowski value, it would be more elegant to simulate a range of models up front in the paper. In particular, the reviewers have requested to see a simulation of the model favoured by the authors (with a separate gist representation) to see how it explains the results of Experiment 2 (e.g., how an outlier is determined by people and how its influence changes with delay).

---

## [Author Response]

Essential revisions:1. As the authors note, a central debate in many fields of cognitive psychology is whether gist is extracted and stored separately from items, or whether only items are stored, and gist is estimated on-the-fly by retrieving multiple items. For example, can memory performance be explained by an exemplar-only model (like Nosofsky's), or is a separate prototype representation needed? I thought that this paradigm offered a neat way to address this debate by comparing the gist reported by participants with the gist estimated from all their item reports. While the two might differ due to random noise, if this difference systematically increased with the retention interval, this would seem to support a separate representation of gist (prototype), whose influence increases with delay/consolidation. However, the authors do not appear to test this simple effect of delay on the difference score between reported gist error (Gr) and estimated gist error (Ge). Is there a reason they didn't do this? They do regress Gr against Ge and Delay, but I'm not sure why regression is better to test the above hypothesis. In any case, the positive relationship between Gr and Ge, but lack of a significant interaction between Ge and Delay, would seem more compatible with an exemplar-only model, i.e., no need for a separate representation of gist. This is where the authors' bias measure may be critical, but I could not intuit whether their finding of increased bias with delay towards Gr (relative to true gist, Gt) rules out an exemplar-only model.

In this comment, and one that follows (comment 7), there is a question about why we did not calculate the error difference or the distance between the reported gist and the estimated gist in any of our analyses. This in some ways appears to be the most obvious way to examine changes in the relation between the two over time, but there is an important problem with that measure that we now explain in the manuscript: Because estimated gist was calculated from an aggregation of item memories which might already be influenced by the reported center after a delay, the estimated gist error (Ge) might have been influenced by the reported center too. Therefore, this analysis would not provide evidence for an item-only model, which assumes participants only remembered the items and the item memories would not be influenced by the gist. If this is the case, the difference between Ge and Gr may remain the same across time because of a persisting dependence of the two (possibly, initially the Gr depends on Ge and over time the Ge starts to depend on Gr). Therefore, we did not think testing the difference between Ge and Gr over time would be helpful to distinguish these hypotheses.

However, your comment inspired us to include a follow-up analysis to contrast a simple form of item-only model with the error data. Critically, as with our baseline bias measure, we assumed that the magnitude of error for each item memory would remain the same, but the direction of error would not be systematically influenced by the gist. In brief, we generated 1000 simulations for each participant. Each simulation consisted of all simulated retrieved items, corresponding to all the landmark locations. For each item location, we randomly generated a retrieved location based on the participant’s true error for this specific item location, allowing angle to vary randomly across the simulations. Then, we computed the center for these locations to get the simulated estimated gist for each simulated participant. The error for such simulated estimated gist was the Euclidean distance between the true center and the simulated estimated gist. The simulated estimated gist error for each real participant was the average value of simulated estimated gist error for their corresponding 1000 simulated participants.

In addition to the results from the analyses described above, we also added Ge as a comparison in the results, as you asked.

There is a significant group interaction for three gist error types *F*(4, 381) = 3.83, *p* <.01, suggesting the increase of three types of gist errors is different over time. For reported gist error and simulated estimated gist error, we found a significant interaction between delay group and gist memory error type, *F*(2, 254) = 6.68, *p* = .001. For estimated gist error and simulated estimated gist error, we also found a significant interaction between delay group and gist memory error type, *F*(2, 254) = 3.28, *p* = .039. Gist error and estimated gist error both increase less over time compared to the simulated estimated gist error under this simple item-only model over time, suggesting that participants’ data are not compatible with this simple item-only model. We did not find a significant interaction between Gr and Ge across time, *F*(2, 254) = 1.18, p = .31. This is consistent with the idea that Ge was calculated from item memories influenced by the center after delay.

There may be other more sophisticated exemplar-based models that can fit our data. However, our results put a constraint (i.e., a center bias) for item-only models, whereas models with a separate gist can readily explain the center bias. We have added a paragraph at the end of Experiment 1 results (page 9, line 195) and included these new results in the supplementary material (page 33, Figure 2-supplement 2). We have also attached the code for these analyses in our most updated code.

2. I think it would really improve the paper if the authors could compare a number of simple computational models, and show the model corresponding to their favoured conclusion (i.e., that a separate gist representation increases its influence on memory over time) is the only one that is consistent with their data. Ideally, this would entail quantitative fitting of parametrised models, but failing that, it might be sufficient to demonstrate qualitatively that certain models can never explain critical results (like the bias measure above, or the effect of an outlier). For example, I presume that an item-only model in which random noise (in both x,y directions) increased with delay could explain their first result of a greater effect of delay on reported item error (Ir) than Gr? But is this model consistent with their regression results, and most importantly, could it never reproduce their bias results?

As discussed in our response to comment 1, we conducted a simulation for the simple item-only model in which random noise increased with delay, and we computed a simulated estimated gist error based on this item-only model (see our response above). Following comment 2, we conducted a regression analysis to evaluate the effect of such simulated estimated gist error on reported gist error. Then, we conducted model comparisons to understand whether reported gist error was best explained by estimated gist error or the simulated gist error based on the item-only model, with delay as an interaction term.

We found that the item-only simulated gist error was associated with gist error (*SSE* = 14248, *F*(1, 124) = 4.81, *p* = .03). However, when the estimated gist error and item-only simulated gist error were both included as predictors, the model did not provide a better fit for the data compared to the model which only has estimated gist error as a predictor (*SSE* = 500.28, *F*(3, 121) = 0.062, *p* = .98). On the other hand, the model with both the estimated gist error and item-only simulated gist error as predictors had a significantly better fit compared to the model which only has item-only simulated gist error as a predictor (*SSE* = 41842, *F*(3, 121) = 5.18, *p* <.01). Taken together, these results suggest that although the item-only model with increased error alone may be associated with an increased gist error, it could not add any explanatory power for the gist error in addition to the estimated gist error.

As for the bias results, the bias in our original submission is a difference score between the absolute bias and baseline bias. The baseline bias is an item-only model in which random noise increased with delay. Our bias results indicate that this simple item-only model could not explain the bias results over time (Figure 3).

3. Another concern is whether the same results would hold if participants' representation of gist (Gr) were more than the simple average of their reported item locations (Ge). For example, would there be any consequences for the authors' conclusions if some 2D locations were represented more accurately than others – e.g., one might expect the location of landmarks close to the edge of the screen to be stored more accurately than ones closer to the centre of the screen (which could be tested by whether edge locations need fewer training trials?). I appreciate that locations are trained to the same criterion, but it is nonetheless possible that some representations are still more precisely encoded than others even after such criterion-training. Then if participants' reported centre were the weighted average of the reported item locations, weighted by each location's remembered precision, could that affect the authors' bias measure? I appreciate that this would not explain why the difference between reported and true centre changes over retention interval, but if one allowed location and precision information to decay at different rates, could this cause such an interaction with retention interval? This could be another model to simulate?

Based on this comment, we first tested whether an estimated center weighting each item by their accuracy will be closer to the center participants reported, compared to the original estimated gist by simple average, using the data from Experiment 1. Each item was given a weight based on its accuracy (1- error/error of all items)/(number of items – 1) such that items with higher accuracy would be weighted more in the estimated gist. We divided each of the weights by the number of items – 1 in order to make the sum of the weight of items equal to 1. We then compared this weighted estimated gist with the estimated gist in their error and the change of their difference in error across time. We found that the two types of error do not differ from each other at 24 hours, 1 week, and 1 month (*p*s >.1). The difference in these two types of error does not change reliably over time (*F*(2, 131) = 1.26, *p* = .28). In addition, we calculated the bias change over time using such weighted estimated gist as center. The bias using the estimated center weighing items by accuracy as bias center increased over time (*SSE* = 2.452, *F*(2, 127) = 11.77, *p* <.001).

Overall, an estimated center that is weighted based on item memory accuracy does not perform differently than an estimated center that weights all items equally. Again, it is possible that other kinds of models weighing items by their accuracy or their remembered precision (i.e., the confidence for the items) could perform differently, or that a study in which item locations were not learned to criterion would result in more variability in item memory accuracy, which could give rise to a gist memory that is meaningfully weighted by item precision. Our study offers a paradigm as a start for future modeling work to explore how the weights of the items can vary according to various kinds of factors, and whether the weights change over time. We have added this point to the discussion (page 17, line 418).

4. A related concern is the more general problem that the mental representations of item locations may not be a simple linear transform of the true locations. If the mental space of locations is a nonlinearly warped version of the physical space (even if still diffeomorphic, e.g., expanded towards the edges of the physical space, rendering edge items more distinct), then not only will the reported item locations (Ir) be systematically biased with respect to their true locations, but the simple average of the reported item locations would not match the reported gist location. Take the simple case of a 1D space with two items encoded at location x=1 and x=3, such that the gist is x=2, but presume the mental representation of those locations were warped by x^2^: then the reported item locations would be 1 and 9 (before downscaling to the original min/max of the space), resulting in an estimated gist (Ge) of 5, whereas the gist in participants' memory (as reported in Gr) would be 2^2^ = 4. I think this potential nonlinear warping of the stimulus space can be captured to some extent by generalising the Euclidean error measure used by the authors to Minkowski's measure (for 2D vectors) = ( (x1-x2) ^g^ + (y1-y2)^g^ ) ^1/g^ (where g=gamma=2 for Euclidean, but could be fit to allow non-Euclidean distances). See for example Tversky and Gati, Psych Review (1982) for further discussion about differences between physical and psychological spaces. When implemented in an item-only model (and the presence of noise that increases with delay), could such nonlinearities/non-Euclidean measures allow such a model to fit the results, particularly the authors' bias measure? Perhaps not, or perhaps the authors can explain such considerations are irrelevant to their results, but I think some thought and ideally modelling about this possibility might be worthwhile.

Thank you for pointing this out. Indeed, the mental representations can be nonlinear. We add this point as well as the paper reference in a discussion paragraph (page 17, line 422). Following your suggestion, we have conducted the same bias analyses with g = 1.5 (as Minkowski’s distance usually has values of g between 1 and 2) and found that this bias change increased over time (*SSE* = 0.86, *F*(2, 131) = 5.419, *p* = .005). This suggests that our result of bias increased over time is not limited to Euclidean distance. However, the result is non exhaustive and other forms of nonlinearities/non-Euclidean measures may allow an item-only model to fit, which may be helpful in explaining the change over time. In addition, even though there may be other item-only models with non-linearly warped space that could explain our change in bias over time, it is not clear whether and how this particular distortion of physical distances would change over time.

**Author response image 2. respfig2:** 

5. The results from the outlier in Experiment 2 lead the authors to say themselves that "the gist influencing item retrieval is not a simple average of the items" – so they clearly need a more complex model in which outliers are down-weighted according to the type of memory question asked, and I wasn't 100% clear what that model was, i.e how exactly the authors explain all the results of both Experiments. I think this is where some simple computational models extended to handle outliers, would be very helpful, in terms of mechanistic understanding.

We conducted a simulation in Experiment 2 comparing a series of models which weight outliers from 0 (not weighted at all) to 1 (over-weighted). We found that the model that generates the estimated center that is closest to the reported center is a model that over-weighted outliers (Results: page 13, line 299; Methods: page 26, line 651). This is a result in comparison to models that down-weighted outliers.

This principle of weighting items that are more distant from the center may apply to Experiment 1 as well. This is a valuable point but unfortunately there is not enough variance across the items on their distance to the center (Mean: 266.9, *SD*: 46) to allow for such analysis and so an aggregated center based on the distance is not much different from a simple average. Future work manipulating the distance of the items towards the center can be done to investigate this principle. We have incorporated this point in the new paragraph illustrating various factors that learners could be weighting items by (page 17, line 423).

6. Regarding the authors' critical bias measure, they write it on page 20 as:"(Encoded Item – Reported Gist) – (Retrieved Item – Reported Gist) / (Encoded Item – Retrieved Item)". Firstly, parentheses around the numerator are missing, i.e. it should be: ((Encoded Item – Reported Gist) – (Retrieved Item – Reported Gist)) / (Encoded Item – Retrieved Item) at least, compared to the code in their bias_func.m, and compared to what would be necessary for the bias to range between 0 and 1. Secondly, if evaluated literally, this measure would always be 1, ((It – Gr) – (Ir – Gr)) = (It – Ir), which is identical to the denominator (where It = true or encoded item locations). I realise this "equation" was meant to give a conceptual idea rather than literal definition, but I think it would be safer to give the precise definition, which is ( d(It,Gr) – d(Ir,Gr) ) / d(It,Ir) where d = some distance function (and It, Ir and Gr are 2D vectors), and in the authors case, they use a Euclidean distance (as in Point 4 above), i.e. d (a,b)=(a1−b1)2+(a2−b2)2.

Thank you for pointing this out. We have edited the methods section in the manuscript (page 21, line 537) to give a precise definition.

7. More importantly, please could the authors help me understand the rationale for this specific bias measure, and why it is so critical for establishing the presence of a gist representation? I appreciate that Gr can be replaced with Ge, but why is this combined measure of item memory better than the simple Gr – Ge gist measure in terms of distinguishing theories? Apologies if I have missed something fundamental to the paper, in which case, perhaps explain a bit more in the Introduction why this particular bias measure is so critical?

This is an important comment that overlaps with comment 1. The Gr – Ge gist measurement will not be able to distinguish the theories for the same reason raised in comment 1. In brief, because Ge was an aggregation of item memories which might already be influenced by the reported center, this analysis would not provide evidence to isolate the influence from the gist on the items.

On the other hand, the specific bias analysis is critical because, by comparing the data with an item-only model baseline, it disentangles error from the bias and shows the magnitude of the particular direction the items were attracted to. We have added a paragraph in the paper in order to clarify this point (page 9, line 195).

8. It is unfortunate that the authors did not counterbalance the order of item and gist memory tests. They do consider this limitation in the Discussion and note that any additive effect of test order would not explain the interaction between memory type and delay, but of course the presence of nonlinear/multiplicative effects (e.g., floor/ceiling effects) means it is not sufficient to conclude that their results could not depend on test order, i.e., it is always better to empirically test generalisation by running the other counterbalancing. This is particularly important here, where there are good a priori, theoretical reasons why test order might matter, e.g., shift people from strategies based on retrieving items or retrieving gist, depending on what type of memory is probed first.

We revised our claim to acknowledge the possibility of nonlinear/multiplicative effects (page 18, line 433).

9. The experimental task provides a somewhat restricted view of gist memory. Thus, it remains unclear what the results mean for other kinds of gist memory, both visual and otherwise. While the authors discuss this point on p.16, memory for gist locations may reflect a combination of basic perceptual strategies (as acknowledged by the authors in their references to ensemble perception) and the demand characteristics of the task. This issue is most prominently seen in the difference between the effects for global and local centres observed in Experiment 2. Here, the instruction to explicitly recall the centre is what may have created (or at least emphasized) the memory for the global centre in the first place. The fact that gist memory was tested first and early on after learning may have changed the associated memory trace, making it difficult to generalize these findings to cases in which gist memories emerge spontaneously. While this may be the price to pay for precise, quantitative error and bias measures of errors, the authors should discuss these limitations in more detail.

We agree with this comment. A sentence reflecting this point has been added to the Discussion section in the discussion paragraph (page 17, line 430).

10. The authors touch on the subject of neural substrates in the Discussion however, I would like to have seen slightly more consideration of this topic. For example, I was surprised by the statement on page: 15 that "neocortex tends to represent information in a 'semanticized' form" – this strikes me as somewhat at odds with more recent theoretical positions on the interplay between episodic and semantic memory (e.g., Renoult et al. TICS 2019) as well the processes by which memory transformation from item to gist memory occurs (Robin and Moscovitch, 2017). While not the focus of this manuscript, it would be helpful for the authors to consider these contemporary positions in relation to their findings.

This is a very good point. We completely agree that our results are not diagnostic of a change in memory system and are indeed consistent with various views on the interplay between episodic and semantic memory. We have rewritten the relevant portion of the discussion accordingly (page 16, line 367).

11. As a point of curiosity, have the authors considered the potential role of individual differences in some of these processes and whether some individuals may be more adept at extracting gist versus item representations?

Very valuable point. We do not have measures of various kinds of cognitive functions that we could use to relate to the variance in learners’ ability to extract the gist versus item representations. We added this point as a future direction in the discussion (page 17, line 430).

12. Is there any relationship of the present results to Brainerd and Reyna's "Fuzzy Trace Theory" (https://en.wikipedia.org/wiki/Fuzzy-trace_theory), which is based on the role of verbatim and gist representations in memory?

According to this theory, memory of a past event has a verbatim copy which is detailed and a fuzzy copy, which is “gisty,” and people rely on the fuzzy copy in making decisions. The meaning of the word “gist” in the present results and fuzzy trace theory differs in the sense that in fuzzy trace theory the word means “imprecise” but in our study it refers to the generalities across items, in particular, a spatial average. This is an important point and we clarified it in the introduction (page 1, footnote 1).

[Editors' note: further revisions were suggested prior to acceptance, as described below.]

Essential revisions:There was some concern regarding the extent to which some of the comments raised by the reviewers had been addressed in the previous revision. The below issues have been raised by the reviewers as warranting serious consideration.(i) The failure to counterbalance the order of gist and item tests was not felt to have been satisfactorily discussed in the Discussion. The reviewers all agree that the paradigm is interesting and relevant, but the lack of counterbalanced replication based on an explicit power analysis limits the generalisability of these findings. The authors need to temper their claims and more deeply acknowledge and address the limitations of this study.

A full paragraph acknowledging and discussing this limitation has been added to the text in the Discussion section (page 20, lines 464-479):

“One limitation of the current experiments is that the testing order (i.e., gist memory before item memory) might have encouraged the retrieval of the items to be consistent with the gist (Tversky and Kahneman, 1974; Mutluturk and Boduroglu, 2014). […] More studies with counterbalanced testing order will be helpful to evaluate this possibility.”

(ii) More emphasis needs to be placed on the modelling, moving the models to the main text and discussing these more fully. The reviewers had suggested simulating a range of models, yet only the simplest of these models was simulated (exemplar model with noise), and this was placed in Supplementary Material. While a weighted version was explored and an alternative Minkowski value, it would be more elegant to simulate a range of models up front in the paper. In particular, the reviewers have requested to see a simulation of the model favoured by the authors (with a separate gist representation) to see how it explains the results of Experiment 2 (e.g., how an outlier is determined by people and how its influence changes with delay).

In our previous submission, we compared our observed data to simulations of bias in a model with noisy item-location representations but no separate gist representation. The reviewer’s suggestion to compare our observed data (and the simulations from an “item only” model) to simulations from an “item plus gist” model is a valuable one, and we have taken some time to consider how best to implement such a model. The complication, of course, is that unlike the selection of an “item only” model, there are an infinite number of “item plus gist” models that we could evaluate, where the degree of influence and the form of the influence of the gist varies. We considered two forms of influence, one in which the gist location influences the probability of a particular retrieved location (instead of the uniform probability distribution around the item location in the “item only model”); and alternatively one in which the gist location shifts the item location (and the surrounding item locations). We ultimately selected the former because it would not change the error of the item memories, making it a closer comparison to the item-only model, and so a version of that “item plus gist” model is now reported in the paper. Turning to the degree of influence, this could be modelled as a parameter in the “item plus gist” model, where increasing influence of gist shifts the probability distribution further from the uniform distribution. The challenge is that, because we did not design the experiment with this sort of modelling exercise in mind, we do not have enough statistical power for a quantitative fitting of parametrized models to determine the best parameter for gist influence per participant (due to the limited number of item retrievals per participant). For the same reason and with no variability in the outlier item, we could not quantitatively examine how the outlier is determined by participants and how its influence changed over time.

We now include a comparison of the observed data to the (original) “item only” model and a new “item plus gist” model (henceforth “gist simulation” for simplicity) as follows: In the gist simulation, the magnitude of error for each item memory is the same as the observed data from participants, but the probability of each location being simulated was inversely related to the distance between that location and the gist (the center) (p25-27, lines 625-664). That is, locations that are closer to the gist were more likely to be retrieved in the simulation compared to locations that are farther from the gist. We computed bias under gist simulation (abbreviated as gist simulated bias) and compared it with the observed bias from participants for both Experiment 1 (methods: p28, lines 675-693) and 2 (methods: p31, lines 764-766). We found that over time, the observed bias became increasingly similar to the gist simulated bias and dissimilar to the item only simulated bias over time in Experiment 1 (as discussed, we could not fit the most accurate parameter of gist influence and therefore the amount of gist influence under the gist simulation is arbitrary and likely not accurately reflecting the amount of gist influence in observed data. Therefore, we did not predict that over time the observed bias would become indistinguishable from the gist simulated bias and focused on testing the change between the difference between the two over time.). In Experiment 2, we observed the same results for the same bias analysis using the local center excluding the outlier (the same analysis is not significant for the global reported center). These results suggest that participants’ bias grew increasingly similar to the gist simulated bias and dissimilar to item only simulated bias over time, suggesting an increasing gist influence on item memories over time. We have added these corresponding results (Experiment 1 results: p8-p10, lines 169-224; Experiment 2 results: p14-16, lines 312-322, 332-343, 352-361) to the main text.

Second, we have added a range of simulations to the main text of Experiment 1, which includes: a bias analysis using the true center of encoded items (rather than the center participants reported), a bias analysis where the center is weighted by item accuracy, and a bias analysis assuming non-Euclidean space (results: “Follow-up bias analyses”, p11, lines 233-261; methods: p28-29, lines 695-708). We found that the results do not differ from our main findings under these different simulations, potentially because the stimuli do not allow for enough variability in item accuray to influence the results.

We know that the efforts we have made to compare our data to different models are not perfect. We hope that the reviewers agree that our results provide a starting point for future endeavours to quantify the gist influence on item memories in long-term memory consolidation and to examine the factors that may affect this influence, such as the variability in item locations and participants’ accuracy and confidence of item memories. We have added these points to the discussion (p21, lines 493-503):

“Our implementation of a gist representation in our bias simulations was very simplistic. […] The current design did not allow for enough variability to tease apart these possibilities, but future research systematically manipulating these factors will be helpful in addressing these issues.”